# Circuit mechanisms underlying embryonic retinal waves

Christiane Voufo[1], Andy Quaen Chen[2], Benjamin E Smith[3], Rongshan Yan[1,2], Marla B Feller[1,2]*, Alexandre Tiriac[1,2]*†

[1]Helen Wills Neuroscience Institute, University of California, Berkeley, Berkeley, United States; [2]Department of Molecular and Cell Biology, University of California, Berkeley, Berkeley, United States; [3]School of Optometry, University of California, Berkeley, Berkeley, United States

**Abstract** Spontaneous activity is a hallmark of developing neural systems. In the retina, spontaneous activity comes in the form of retinal waves, comprised of three stages persisting from embryonic day 16 (E16) to eye opening at postnatal day 14 (P14). Though postnatal retinal waves have been well characterized, little is known about the spatiotemporal properties or the mechanisms mediating embryonic retinal waves, designated stage 1 waves. Using a custom-built macroscope to record spontaneous calcium transients from whole embryonic retinas, we show that stage 1 waves are initiated at several locations across the retina and propagate across a broad range of areas. Blocking gap junctions reduced the frequency and size of stage 1 waves, nearly abolishing them. Global blockade of nAChRs similarly nearly abolished stage 1 waves. Thus, stage 1 waves are mediated by a complex circuitry involving subtypes of nAChRs and gap junctions. Stage 1 waves in mice lacking the β2 subunit of the nAChRs (β2-nAChR-KO) persisted with altered propagation properties and were abolished by a gap junction blocker. To assay the impact of stage 1 waves on retinal development, we compared the spatial distribution of a subtype of retinal ganglion cells, intrinsically photosensitive retinal ganglion cells (ipRGCs), which undergo a significant amount of cell death, in WT and β2-nAChR-KO mice. We found that the developmental decrease in ipRGC density is preserved between WT and β2-nAChR-KO mice, indicating that processes regulating ipRGC numbers and distributions are not influenced by spontaneous activity.

**\*For correspondence:**
mfeller@berkeley.edu (MBF);
alexandre.tiriac@vanderbilt.edu
(AT)

**Present address:** †Biological
Sciences and Vanderbilt Brain
Institute, Vanderbilt University,
Nashville, United States

**Competing interest:** See page
17

**Reviewing Editor:** Fred Rieke,
University of Washington, United
States

## Editor's evaluation

This paper investigates waves in embryonic mouse retinas. These stage 1 waves have been studied less than the post-natal (stage 2) waves. The paper uses elegant imaging and analysis approaches to monitor calcium signals across the entire retina and to determine the properties of the waves and their dependence on cholinergic and electrical synapses. This contributes substantially to the understanding of how these waves are generated.

## Introduction

Throughout the developing nervous system, spontaneous activity is observed before neural circuits are fully formed and sensory transduction begins (*Akin and Zipursky, 2020*; *Blankenship and Feller, 2010*; *Luhmann and Khazipov, 2018*; *Martini et al., 2021*). This activity is implicated in several development events, including cell death, maturation of functional circuits, and refinement of axonal projections (*Blanquie et al., 2017a*; *Fujimoto et al., 2019*; *Kirkby et al., 2013*). This is well studied in the developing visual system, where prior to the maturation of vision, laterally propagating spontaneous depolarizations sweep across retinal ganglion cells (RGCs), a pattern referred to as retinal

waves (*Wong, 1999*). Retinal waves drive eye-specific segregation and retinotopic refinement of retinal projections to the dorsal lateral geniculate nucleus and superior colliculus (*Ackman and Crair, 2014*; *Arroyo and Feller, 2016*). Retinal waves also play a role in the maturation of direction-selectivity within the retina (*Tiriac et al., 2022*) and superior colliculus (*Ge et al., 2021*; *Wang et al., 2009*), as well as the development of retinal vasculature (*Biswas et al., 2020*; *Weiner et al., 2019*).

Retinal waves are present throughout mouse retinal development, starting as early as embryonic day 16 and persisting until postnatal day 14, which is around the time of eye opening (*Blankenship and Feller, 2010*; *Choi et al., 2021*; *Feller and Kerschensteiner, 2013*). As the retina develops, the circuits that mediate waves change. Stage 2 retinal waves, observed between postnatal day 1 and 10 (P1–10), are mediated via activation of nicotinic acetylcholine receptors (nAChRs) by acetylcholine (ACh) released from starburst amacrine cells (SACs). Stage 3 retinal waves, observed between P10 and 14, are mediated via activation of ionotropic glutamate receptors by glutamate released from bipolar cells.

Stage 1 waves are perhaps the least well understood, yet they are concurrent with many important events in retinal development as well as with retinal projections reaching their targets in the brain (*Martini et al., 2021*). Stage 1 waves are present in mice between embryonic day 16 and 18 (E16–18) (*Bansal et al., 2000*) and in rabbit starting at E22 (*Syed et al., 2004*). In rabbit, stage 1 waves persist in the presence of pharmacological antagonists of fast neurotransmitters and are blocked by gap junction antagonists (*Syed et al., 2004*). In mice, stage 1 waves consist of large propagating waves and small non-propagating events (*Bansal et al., 2000*). The application of nAChR antagonist inhibits larger propagating waves (*Bansal et al., 2000*). Though there is no anatomical evidence of synapses as early as E16–18, recent work has shown that SACs are present embryonically, begin to migrate to the inner nuclear layer (INL), and send projections to the inner plexiform layer (IPL) guided by homotypic contacts (*Ray et al., 2018*). Hence, cholinergic signaling is likely occurring via the volumetric release of ACh (*Ford et al., 2012*). The relative contribution of gap junction coupling and cholinergic signaling to the spatiotemporal properties of stage 1 waves remains to be understood.

Here we describe the spatiotemporal properties of stage 1 waves across the whole retina using a novel macroscope. We then identify the role of gap junction and cholinergic circuits on the generation and propagation of stage 1 waves. Next, we explore stage 1 waves in the β2-nAChR-KO mouse, which is the canonical mouse model for studying the role of stage 2 waves in developmental processes, and report that this mouse also exhibits altered stage 1 activity. Finally, we use β2-nAChR-KO mice to demonstrate that the regulation of ipRGC density is a wave-independent process.

## Results

### Macroscope imaging reveals the spatiotemporal properties of stage 1 retinal waves

The mouse retina at E16–18 exhibits spontaneous correlated transients (*Bansal et al., 2000*; *Syed et al., 2004*), termed stage 1 retinal waves, despite the immature state of retinal circuits. At E16–18, several postmitotic cell types are present in the retina, including broad classes of retinal ganglion cells (RGCs) and amacrine cells (ACs), as well as proliferating progenitors which will go on to produce cells such as rods, bipolar cells, and Muller glia postnatally (*Figure 1A*; *Cepko, 2014*). By E17, there are no chemical synaptic structures (*Hoon et al., 2014*), though migrating SACs release ACh (*Wong, 1995*), and they, along with RGCs, express nicotinic acetylcholine receptors. One potential mode of cell–cell communication is via gap junction coupling between RGCs as well as between progenitor cells (*Cook and Becker, 2009*), which have been proposed to be the primary substrate mediating stage 1 waves (*Catsicas et al., 1998*; *Syed et al., 2004*; *Wong et al., 1998*).

To begin to understand how cholinergic signaling and gap junction coupling govern stage 1 waves, we first used calcium imaging to characterize their spatiotemporal properties. Retinas were isolated from E16–18 mice that were either bath loaded with the organic calcium dye Cal 520 or from mice expressing the genetically encoded calcium indicator GCaMP6s under the *Vglut2* promoter (*Vglut2;G-CaMP6s*). The earliest age at which we could detect reliable waves was E16 (*Video 1*). Stage 1 retinal waves were recorded on retinal whole mounts using a custom-built epifluorescent macroscope.

To assess the spatiotemporal properties of stage 1 waves, we divided the retinal surface into small square ROIs (roughly 10 μm × 10 μm), about 7 μm apart. ΔF/F traces for each ROI were rasterized

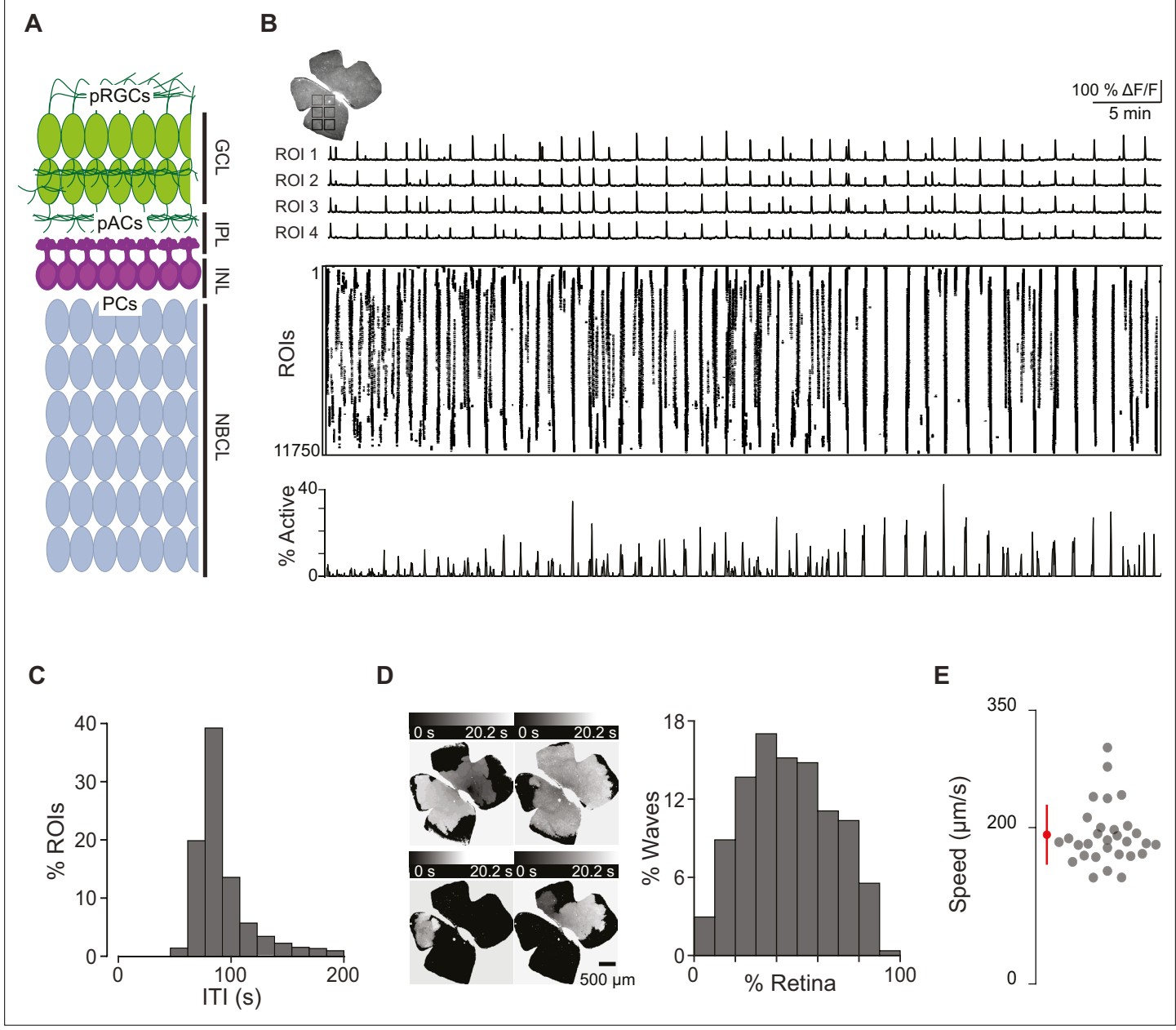

**Figure 1.** Spatiotemporal characteristics of embryonic waves. (**A**) Schematized cross section of an E16 retina when stage 1 waves begin. Green cells represent post-mitotic retinal ganglion cells (pRGCs). Magenta cells represent post-mitotic amacrine cells (pACs). Gray cells represent progenitor cells (PCs). GCL, ganglion cell layer; IPL, inner plexiform layer; INL, inner nuclear layer; NBCL, neuroblastic cell layer. (**B**) Top left: macroscope image of the baseline fluorescence of an E17 GCaMP6s retina. For subsequent analysis, the retina was divided into 11,750 10 μm × 10 μm (not drawn to scale) squares. Number of ROIs changed depending on retina size. Top: ΔF/F traces of calcium transients from four example 10 μm × 10 μm ROIs. Middle: rasterized calcium transients for all ROIs raster plot of calcium transients >50% ΔF/F. Bottom: percentage of ROIs active throughout the time course of the recording (1 hr). (**C**) Histogram showing the distribution of the percent of ROIs with inter-transient-intervals (ITIs) ranging from 0 to 200 s. (**D**) Left: heatmap showing the temporal progression and spread of four example waves. Scale depicts timescale of propagation; dark gray = start of propagation and white = end of propagation. Right: histogram showing the distribution of waves with an area ranging from 0 to 100% of the retina. (**E**) Summary plot of wave speeds. Red dot and line = mean and standard deviation, respectively. n = 30 waves, three retinas, three mice.

The online version of this article includes the following source data and figure supplement(s) for figure 1:

**Source data 1.** Inter-transient-interval, wave area, and wave speed.

**Figure supplement 1.** Distribution of stage 1 wave sizes.

**Figure supplement 2.** Average wave speed measurement.

**Figure supplement 3.** Distribution of stage 1 wave initiation sites.

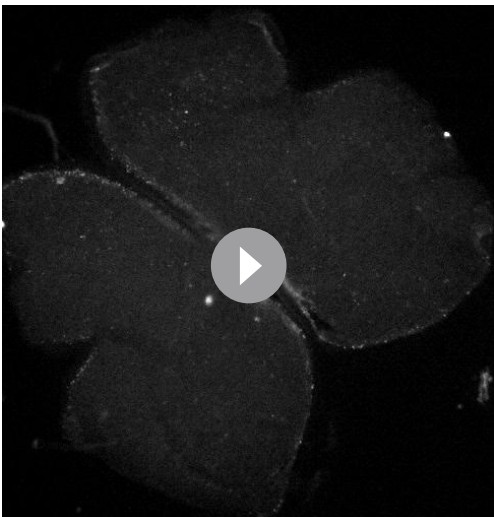 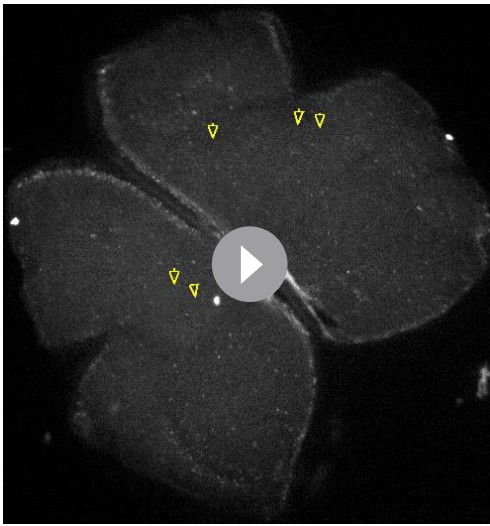

**Video 1.** Stage 1 waves. Calcium imaging of stage 1 waves using the macroscope. Total field of view is 4.7 mm × 4.7 mm. Frame rate 10 Hz. Total length of movie represents 1 min of recording.

https://elifesciences.org/articles/81983/figures#video1

**Video 2.** Example of small non-propagating events. Calcium imaging of stage 1 waves using the macroscope. Total field of view is 4.7 mm × 4.7 mm. Frame rate 10 Hz. Total length of movie represents 15 s of recording. Yellow triangles depict small non-propagating events.

https://elifesciences.org/articles/81983/figures#video2

based on a detection algorithm that identified the timing of peak changes in fluorescence, which we call transients (*Figure 1B*). These rasterized transients were used for subsequent analysis. We first computed the time between spontaneous transients by measuring the inter-transient interval (ITI) for each ROI and found that the distribution peaked at around 80 s (*Figure 1C*). We also described the size of individual waves by computing the sum of transients that occurred simultaneously (i.e., percent of ROIs that participated in each wave). The distribution of wave sizes was broad, ranging from 6 to 90% of the retina, with a mean and standard deviation of 46 ± 21% (*Figure 1D*).

In addition to propagating waves, we also observed small non-propagating calcium events (defined by transients present in a few neighboring ROIs that never propagated beyond 160 × 160 µm²; *Video 2* and *Figure 1—figure supplement 1*). These events were like those described previously in *Bansal et al., 2000*. Note that these small non-propagating transients were included in the ITI but not in the analysis of propagating wave sizes.

Finally, we calculated the average propagation speed of stage 1 waves to range from 145 µm/s to 237 µm/s (average ± SD: 181 ± 24 µm/s; *Figure 1E*, *Figure 1—figure supplement 2*), similar to stage 2 waves we observed with an average speed of 177 µm/s ± 62 µm/s (see *Table 1* for wave speed summary data). We also measured the distribution of stage 1 wave initiation sites in WT retinas and saw no evidence of an initiation site bias (*Figure 1—figure supplement 3*).

## nAChRs and gap junctions are important for setting the frequency and area of stage 1 waves

Previous work based on epifluorescent calcium imaging experiments performed in embryonic mice has shown that retinal waves, as defined by correlated changes in fluorescence, are reduced in frequency and size by curare, a competitive antagonist for nAChRs (*Bansal et al., 2000*). However, in roughly the equivalent developmental period in rabbit, blockade of all fast neurotransmitter receptor, including nAChRs, had no impact on wave frequency (*Syed et al., 2004*). Rather, waves in rabbit are blocked after the application of 18β-glycyrrhetinic acid, a gap junction antagonist (*Syed et al., 2004*).

To determine the relative role of gap junctions and nAChRs on the frequency and area of stage 1 waves in mice, we used two-photon calcium imaging and pharmacology in retinas isolated from E16–18 mice bath loaded with Cal 520. To block gap junctions, we bath applied the gap junction antagonist meclofenamic acid (MFA, 50 µM), which reversibly blocks electrical coupling between retinal interneurons (*Veruki and Hartveit, 2009*) and developing ipRGCs (*Caval-Holme et al., 2019*).

**Table 1.** Stage 1 wave speeds in WT and β2-nAChR-KO, plus stage 2 wave speeds.
Speeds recorded from individual waves. Summary data (µm/s) reported in *Figure 3* with additional data to compare with speeds of stage 2 waves. Each row is an individual wave.

| | WT stage 1 waves | B2-nAChR-KO stage 1 waves | Stage 2 waves |
|---|---|---|---|
| | 174.17 | 77.91 | 205.68 |
| | 183.94 | 92.40 | 124.44 |
| | 192.19 | 98.19 | 153.25 |
| | 162.26 | 117.34 | 120.47 |
| | 189.84 | 128.52 | 158.59 |
| | 237.34 | 87.97 | 176.49 |
| | 179.24 | 92.40 | 263.86 |
| | 165.68 | 92.06 | 156.09 |
| | 144.52 | 103.25 | 108.62 |
| | 182.07 | 103.90 | 299.29 |
| Avg | 181.12 | 99.39 | 176.68 |
| Stdev | 24.39 | 14.72 | 62.62 |

Application of the gap junction blocker MFA (50 µM) nearly abolished stage 1 waves, causing a significant reduction in frequency of waves and cell participation during waves (*Figure 2A-C*). We also found a small but significant reduction in wave amplitude, quantified as the average maximum response amplitude of all cells participating in individual waves (*Figure 2D*).

MFA has notable off-target effects including overall cell health after long exposures (*Kuo et al., 2016*). Previous studies from our lab show that at P6 MFA does not reduce light-induced calcium transients in M1-iRGCs, the ability of M4-ipRGCs to fire action potentials, nor the amplitude of depolarization-induced calcium transients (*Caval-Holme et al., 2019*). Note this is in contrast to another gap junction antagonist, carbenoxolone, which inhibits light responses in cultured ipRGCs (*Bramley et al., 2011*). Using whole-cell voltage-clamp recordings, we observed that MFA induced a significant decrease in voltage-activated potassium conductance (*Figure 2—figure supplement 1*). Note that a reduction in K+ conductance would increase excitability of cells and is therefore unlikely to be the reason for the observed decrease in wave activity in the presence of MFA. Hence, we conclude that MFA's block on retinal waves is via their impact on gap junctions.

We next assayed the impact on nAChR-antagonists on stage 1 waves. For stage 2 waves, both spontaneous calcium transients and compound excitatory synaptic events are completely blocked by bath application of dihydro-ß-erythroidine hydrobromide (DHβE, 8 µM) (*Ford et al., 2012*), which preferentially targets nAChRs containing α4 and β2-subunits (*Harvey and Luetje, 1996*; *Harvey et al., 1996*). We found that DHβE also dramatically reduced activity associated with stage 1 retinal waves (*Figure 2E-H*). However, in contrast to stage 2, some waves persisted in the presence of DHβE, but recruited fewer neurons and therefore had smaller areas (*Figure 2G*). Stage 1 waves were blocked by general nAChR antagonists: both hexamethonium (*Figure 2I-J*; Hex, 100 µM), a non-selective nAChR antagonist, and epibatidine (*Figure 2K-L*; EPB, 10 nM), an nAChR agonist that potently desensitizes all nAChRs (*Corrie et al., 2020*; *Spang et al., 2000*), blocked stage 1 wave events. Hence, both gap junctions and multiple subunit combinations of nAChRs mediate the initiation and propagation of stage 1 waves .

## β2-nAChR knock-out mice exhibit perturbed stage 1 waves

Our results thus far indicate that both the frequency and area of stage 1 retinal waves are modulated by the activation of different subtypes of nAChRs as well as gap junction coupling. To further differentiate the role of different nAChRs, we characterized mice where the β2 subunit of the nicotinic acetylcholine receptor is genetically ablated (β2-nAChR-KO). β2-nAChR-KO mice have severely disrupted stage 2 retinal waves (*Bansal et al., 2000*; *Rossi et al., 2001*) and have served as a model system for

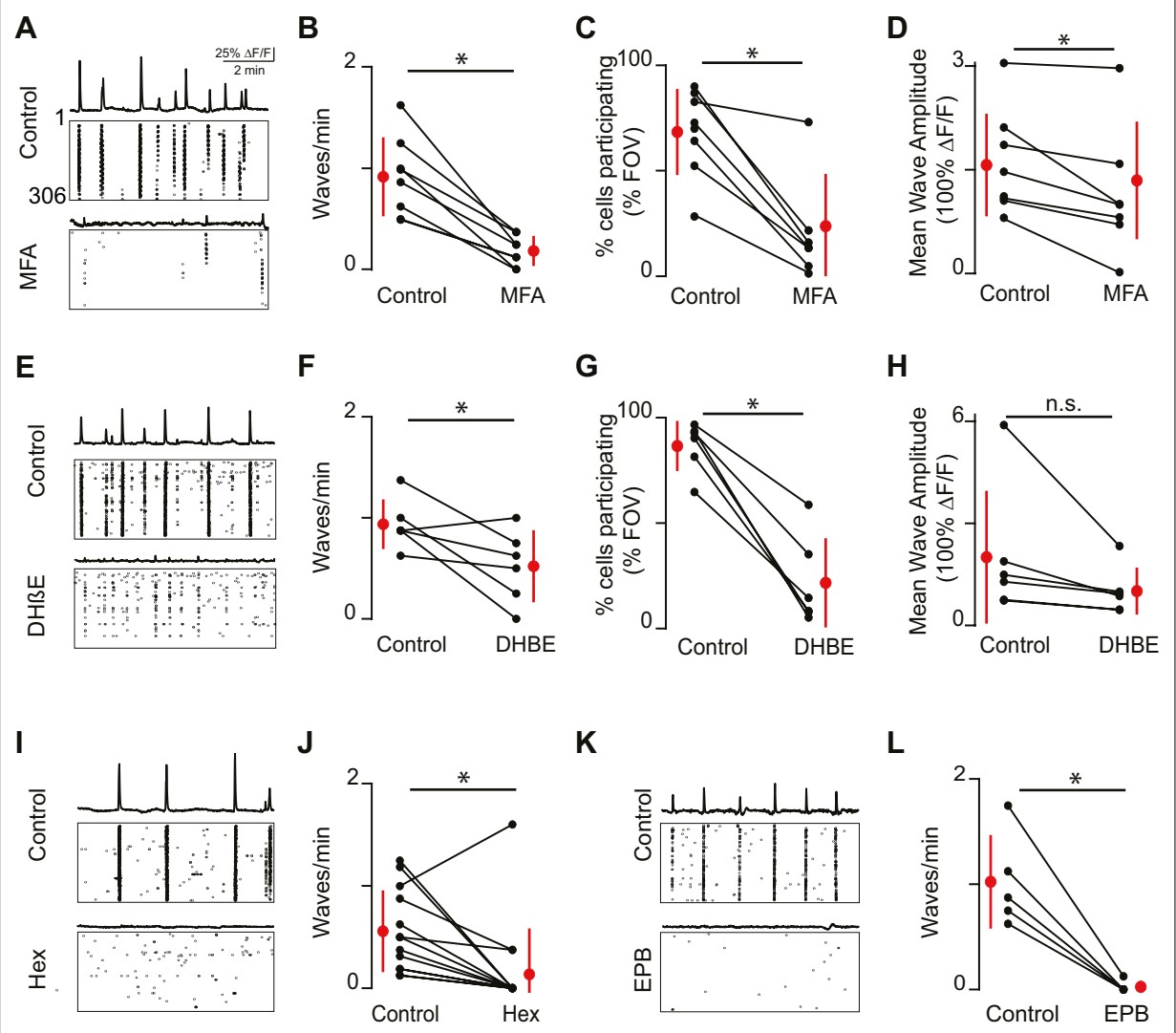

**Figure 2.** Embryonic waves are mediated by gap junction and cholinergic circuits. (**A**) ΔF/F time course of spontaneous activity observed in the field of view (FOV) and raster plot of neuronal calcium transients in ACSF (control condition, top) and in the presence of meclofenamic acid (MFA) (bottom). (**B**) Summary plot showing frequency of waves in control and MFA. Red dots and lines = mean and standard deviation, respectively. Asterisks represents significant effects. n = 8 retinas (six mice); p=4.81e−4. (**C**) Percent cells that participate in retinal waves in control and MFA (50 µM) (p=3e−3). (**D**). Mean calcium response of the neurons that participate in waves in control and MFA (p=0.02). (**E–H**) Same as (**A–D**) for dihydro-β-erythroidine (DHβE, 8 µM). (**F–H**) (n = 6 retinas; six mice); (**F**) p=0.05; (**G**) p=4.58e−4; (**H**) p=0.09. (**I, J**) Same as (**A, B**) but with hexamethonium (Hex, 100 µM) following a baseline recording. n = 14 retinas (nine mice); p=1.48e−2. (**K–L**) Same as (**A, B**) but with epibatidine (EPB, 10 nM) following a baseline recording. n = 5 retinas (five mice); p=4.8e−3. All statistical tests here are paired t-tests.

The online version of this article includes the following source data and figure supplement(s) for figure 2:

**Source data 1.** Event frequency.

**Figure supplement 1.** Controls for off-target effects of meclofenamic acid (MFA) on E16–18 retinal ganglion cells (RGCs).

assessing the role of stage 2 waves in driving different developmental events (*Ackman et al., 2012*; *Burbridge et al., 2014*).

We observed that β2-nAChR-KO retinas exhibited stage 1 waves with different spatiotemporal properties than those of WT retinas (*Figure 3A and B*, see also *Video 3*). Specifically, β2-nAChR-KO retinas exhibited longer IEIs (*Figure 3C and D*), and individual waves propagated over smaller areas β2-nAChR-KO (*Figure 3E* and *Figure 1—figure supplement 1*). In sharp contrast to WT retina, waves in the β2-nAChR-KO retina were unaffected by the addition of Hex but showed a significant reduction in both area and frequency with the application of MFA (*Figure 3D and E*). These results suggest that

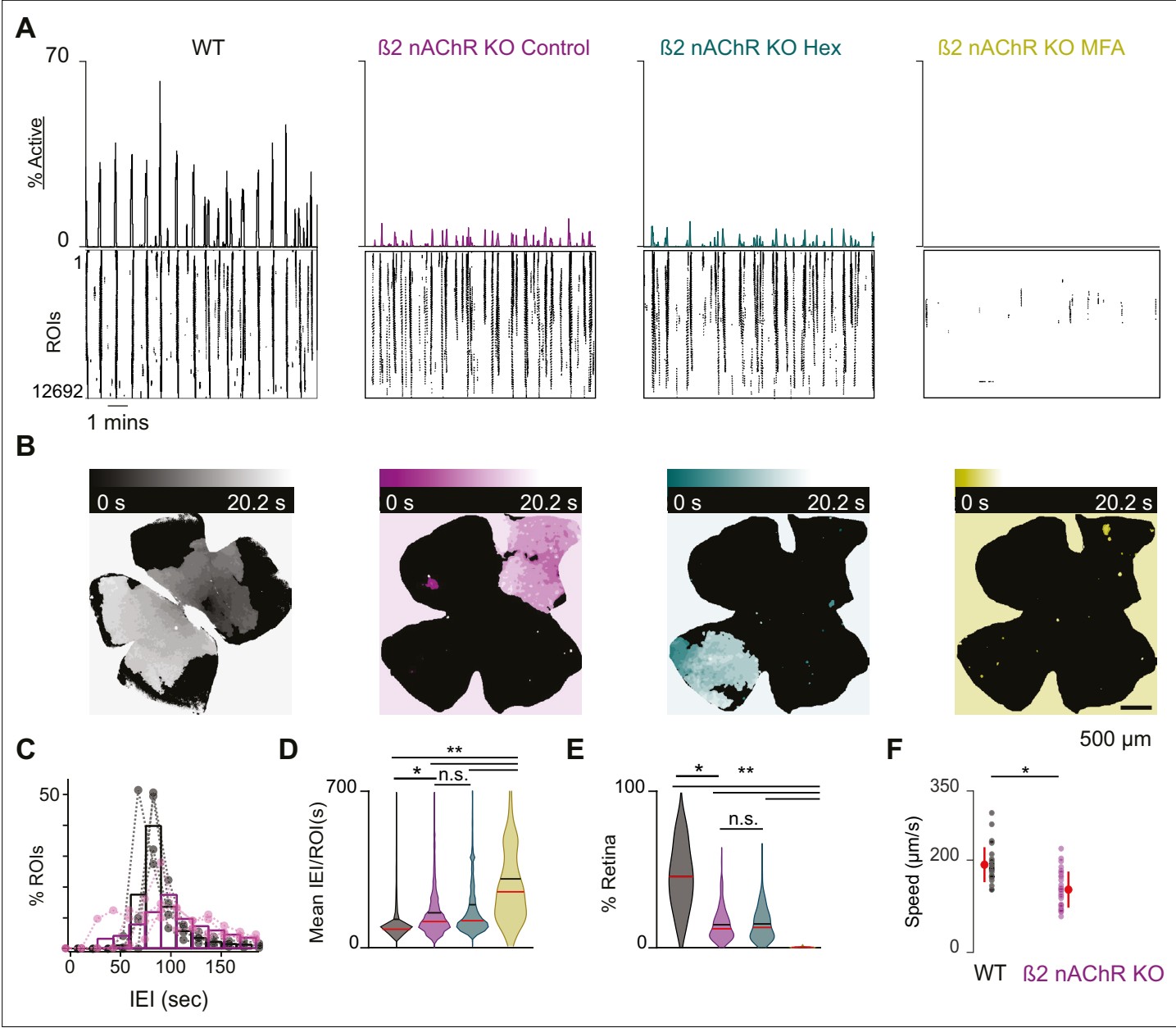

**Figure 3.** β2-nAChR-KO mice have reduced stage 1 wave activity. (**A**) Top: area plot summarizing the percentage of ROIs active throughout the time course of the recording; bottom: raster plots of ROI calcium transients across conditions. Black: WT control; magenta: β2-nAChR-KO control; teal: β2-nAChR-KO in Hex; yellow: β2-nAChR-KO in meclofenamic acid (MFA). (**B**) Heatmap showing the temporal progression of a propagating event observed using epifluorescent calcium imaging on the macroscope across experimental conditions. Scale depicts timescale of propagation; dark colors = start of propagation; white = end of propagation. (**C**) Histogram showing the distribution of the percent of ROIs with inter-transient-intervals (ITIs) ranging from 0 to 200 s. Bars represent summary data across all retinas. Black bars are same data as *Figure 1C*. Dots represent distributions for individual retinas. mean IEI/ROI in WT and β2-nAChR-KO retinas in control conditions. (D-E) Violin plots summarizing the distribution of mean IEI/ROI (D) and event area (E) across experimental conditions. n = 4 retinas, four mice (WT), n = 5 retinas, five mice (β2-nAChR-KO). Black bar = mean; red bar = median. **p<0.01, *p<0.05. One-way ANOVA, followed by Tukey–Kramer post hoc test . (**F**) Summary plot of wave speeds. WT are the same data as *Figure 1E*. Red dot and line = mean and standard deviation, respectively. n = 30 waves, three retinas, three mice per condition p=1.19e$^{-6}$. Unpaired t-test.

The online version of this article includes the following source data and figure supplement(s) for figure 3:

**Source data 1.** Inter-transient-interval, wave area, and wave speed: WT vs. β2-nAChRKO.

**Figure supplement 1.** Frequency of stage 1 waves in WT and β2-nAChR-KO retinas recorded on the macroscope vs. the two-photon microscope.

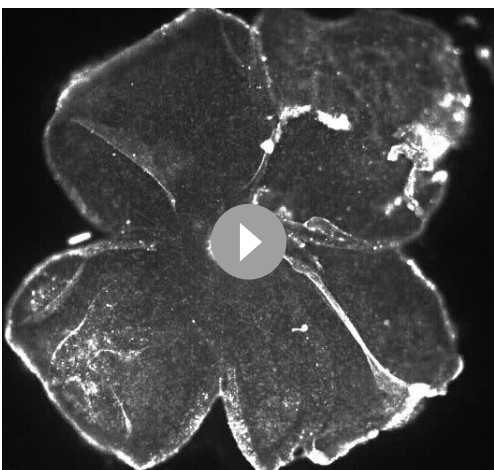

**Video 3.** Stage 1 waves in β2-nAChR-Kos. Calcium imaging of stage 1 waves in retinas isolated from β2-nAChR-KO mice using macroscope. Total field of view is 4.7 mm × 4.7 mm. Frame rate 10 Hz. Total length of movie represents 1 min of recording.

https://elifesciences.org/articles/81983/figures#video3

in the absence of β2-nAChRs, gap junctions are the primary remaining component of the stage 1 wave generation mechanism. Furthermore, we found that stage 1 waves in the β2-nAChR-KO propagated at a significantly slower speed than those observed in WT mice (average ± SD: 99 ± 15 µm/s compared to 181 ± 24 µm/s for WT, n = 10 waves per condition, p=3.9073e$^{-8}$) (*Figure 3F*), consistent with a distinct mechanism of propagation.

Finally, we wanted to test whether the properties of stage 1 waves in WT or β2-nAChR-KO were influenced by light activation of ipRGCs via the 474 nm imaging light used to excite the calcium dye on the macroscope. ipRGCs have been shown to be responsive to 476 nm light during both embryonic and postnatal development (*Emanuel and Do, 2015*; *Verweij et al., 2019*). Previous studies have shown a light-dependent increase in the wave frequency of stage 2 waves in the β2-nAChR-KO during the first postnatal week (*Kirkby and Feller, 2013*). This light modulation of the spatiotemporal properties of β2-nAChR-KO during the first postnatal week depends on ipRGC melanopsin expression, as well as an increase in gap junction conductance between ipRGCs and other RGCs (*Arroyo et al., 2016*; *Kirkby and Feller, 2013*). However, we found no difference in the frequency of waves recorded using a two-photon microscope (based on 920 nm illumination) to those recorded on the macroscope and in either the WT or β2 nAChR-KO embryonic retinas (*Figure 3* and *Figure 3—figure supplement 1*). Hence, we conclude that light activation of ipRGCs does not significantly influence the spatiotemporal properties of stage 1 retinal waves in either WT or the β2 nAChR-KO retinas. These results are consistent with the fact that light stimulation of the retina does not modulate the frequency of stage 2 waves and only begins to do so when conventional photoreceptors come online during stage 3 waves (*Tiriac et al., 2018*, *Renna et al., 2011*).

## ipRGCs participate in stage 1 waves but their density and distribution are not altered in the β2-nAChRs-KO

RGCs undergo a period of dramatic cell death during the first two postnatal weeks of development, the majority occurring during the first postnatal week (*Abed et al., 2022*; *Braunger et al., 2014*). Whether this cell death process is regulated by retinal waves is unknown. We looked specifically at intrinsically photosensitive ganglion cells (ipRGCs) for several reasons. First, ipRGCs have completed proliferation (*Lucas and Schmidt, 2019*; *McNeill et al., 2011*) and appear to be fully differentiated by E16 (*Shekhar et al., 2022*; *Whitney et al., 2022*), the onset of stage 1 waves. ipRGCs undergo a period of dramatic cell death during the first two postnatal weeks of development, the majority occurring during the first postnatal week. Prevention of cell death profoundly disrupts several important developmental processes in the retina – including spacing of ipRGC somas as well as rod- and cone-mediated circadian entrainment through the activation of ipRGCs (*Chen et al., 2013*). However, the exact mechanism regulating ipRGC cell death is unknown. Here we assessed the impact of disrupting stage 1 and 2 waves on the density and distribution of ipRGCs.

We first set out to determine whether ipRGCs participate in stage 1 waves. To do this, we conducted two-photon calcium imaging of RGCs in the ganglion cell layer (GCL) of retinas isolated from *Opn4$^{Cre/+}$;Ai9* E16–18 mice (*Ecker et al., 2010*), which express tdTomato in ipRGCs (*Figure 4A*) enabling us to assess the differential participation of RGCs and ipRGCs during stage 1 waves (*Figure 4B*). On average, both RGCs and ipRGCs participated in most waves with no significant differences between the two groups (*Figure 4C*; average ± SD: RGCs 78.78 ± 21.48%; ipRGCs 84.64 ± 16.86%). We also found no significant differences in the amplitude of the calcium response that RGCs

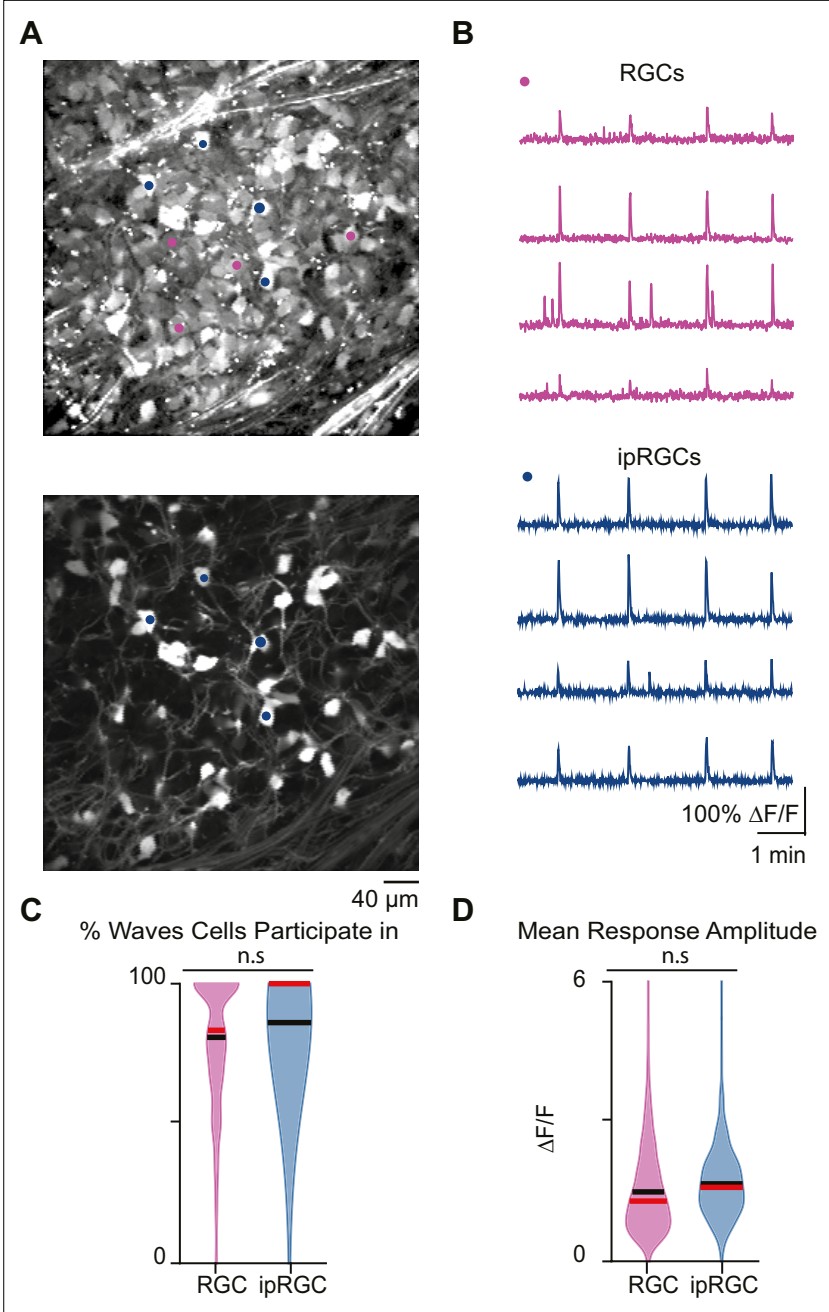

**Figure 4.** Stage 1 waves robustly recruit intrinsically photosensitive ganglion cells (ipRGCs) and the general retinal ganglion cell (RGC) population. (**A**) Top: example field of view (FOV) where pixel intensity was averaged across all frames (712) to get representative image of Cal 520 bath-loaded RGCs. Pink dots correspond to four example RGCs and blue dots correspond to four example ipRGCs Bottom: same example FOV from the top image, with tdTom signal averaged across all frames. Blue dots the same as those in the top image. (**B**) Top: traces of the four example RGCs marked by the pink circles in (**A**). Bottom: traces of four example ipRGCs marked by the blue circles in (**A**). (**C**) Violin plot of percentage of waves each cell participated in. n = 20 retinas (13 mice); p=0.09. Unpaired t-test. (**D**) Violin plot of mean event amplitude/cell/FOV. n = 20 retinas (13 mice); p=0.06. Unpaired t-test. Black bar = mean; red bar = median.

The online version of this article includes the following source data for figure 4:

**Source data 1.** Intrinsically photosensitive ganglion cell (ipRGC) participation and mean response amplitude.

and ipRGCs exhibit in response to stage 1 waves (*Figure 4D*). Hence, ipRGCs are depolarized by stage 1 waves similarly to stage 2 waves (*Caval-Holme et al., 2019*; *Caval-Holme et al., 2022*; *Chew et al., 2017*).

Since β2-nAChR-KO retinas exhibited reduced retinal activity during both embryonic and early postnatal development, we used this mouse as a model to determine whether normal stage 1 and 2 wave activity is important for regulating the density and distribution of ipRGCs. To this end, we isolated retinas from *Opn4$^{Cre/+}$;Ai9* and *Opn4$^{Cre/+}$;Ai9;(β2-nAChR$^{-/-}$)* at P1 and P7. These retinas express tdTomato in all melanopsin-expressing cells regardless of subtype. Retinas were imaged using the macroscope (*Figure 5A and B*).

We observed a dramatic decrease in the density of ipRGCs in WT retinas from P1 to P7 (*Figure 5B and C*; p=1.17 × 10$^{-7}$), consistent with previous studies of ipRGCs (*Chen et al., 2013*) and which coincides with peak levels of RGC apoptosis, the primary cause of RGC death during development (*Braunger et al., 2014*; *Young, 1984*). We found that β2-nAChR-KO retinas exhibited the same density of ipRGCs at P1 as WT retinas, suggesting that the decrease in activity in stage 1 waves does not regulate ipRGC cell density. Like WTs, β2-nAChR-KO retinas also exhibited a decrease in the density of ipRGCs from P1 to P7 (p=4.99 × 10$^{-11}$). At P7, we observed a small but significant increase in ipRGC densities at P7 in β2-nAChR-KO mice than in WT mice (568 ± 33 ipRGCs/µm$^2$ in WT vs. 654 ± 78 ipRGCs/µm$^2$ in β2-nAChR-KO, n = 8 retinas in each genotype). We cannot determine whether this small difference is due to the smaller size of retinas in β2-nAChR-KO retinas (*Xu et al., 1999*) or reflects a true increase in cell number. Overall, these data indicate that the cell death processes that regulate ipRGC density during the first postnatal week persist despite a significant reduction in wave activity.

To determine the impact of this developmental decrease in cell density on the mosaic organization of ipRGCs, we computed the regularity index, which is equal to the average nearest-neighbor distance divided by the standard deviation. A large regularity index is associated with a non-random distribution of somas. Despite the expected increase in nearest-neighbor distance in WT and β2-nAChR-KOs retina between P1 and P7 (*Figure 5D*, see *Table 2*), there was only a small decrease in the regularity index. Interestingly, the measured mean regularity indices of 2.8 ± 0.12 (WT P1), 2.9 ± 0.16 (β2-nAChR-KO P1), 2.3 ± 0.13 (WT P7) and 2.6 ± 0.2 (β2-nAChR-KO P1) fall within the range of what would be predicted by a random distribution of cells with soma diameters between 7–10 µm (*Keeley et al., 2020*). Hence, the decrease in cell density does not appear make the soma organization more ordered. This might be expected since our analysis does not differentiate between ipRGC subtypes, each of which likely forms an independent retinal mosaic. Together these data indicate that although retinal waves provide a robust source of depolarization for embryonic and early postnatal ipRGCs, reducing wave activity does not significantly influence the processes that regulate their density or organization. Whether elimination of all spontaneous activity affects these processes remains to be determined.

## Discussion

We show that stage 1 retinal waves are a robust source of spontaneous activity in the embryonic retina. Stage 1 waves initiate throughout the retina, propagate over finite regions of varying size, and drive periodic depolarizations of neurons in the immature retinal GCL. In WT embryonic mice, stage 1 waves are abolished in the presence of general nAChR antagonists but persist, albeit with greatly reduced cell participation, in an nAChR antagonist that targets α4β2 containing nAChRs. In the presence of gap junction antagonists, the frequency and participation of cells within waves is also greatly diminished, though some waves still occur. We found that in retinas isolated from embryonic β2-nAChR-KO mice, which exhibit strongly reduced stage 2 wave frequency, waves persisted both in control conditions and in the presence of a general nAChR antagonist, though the area of waves is greatly diminished compared to WT. However, similar to what we observed in WT mice, the frequency of stage 1 waves in β2-nAChR-KO mice was greatly reduced in the presence of a gap junction antagonist. This more striking effect of the gap junction antagonist compared to the general nAChR antagonist indicates that the electrical synapses are sufficient to generate waves in β2-nAChR-KO mice. Finally, we showed that ipRGCs are depolarized by stage 1 waves, but that the decrease in density of ipRGCs across early postnatal development was unaffected in the β2-nAChR-KO mouse, indicating

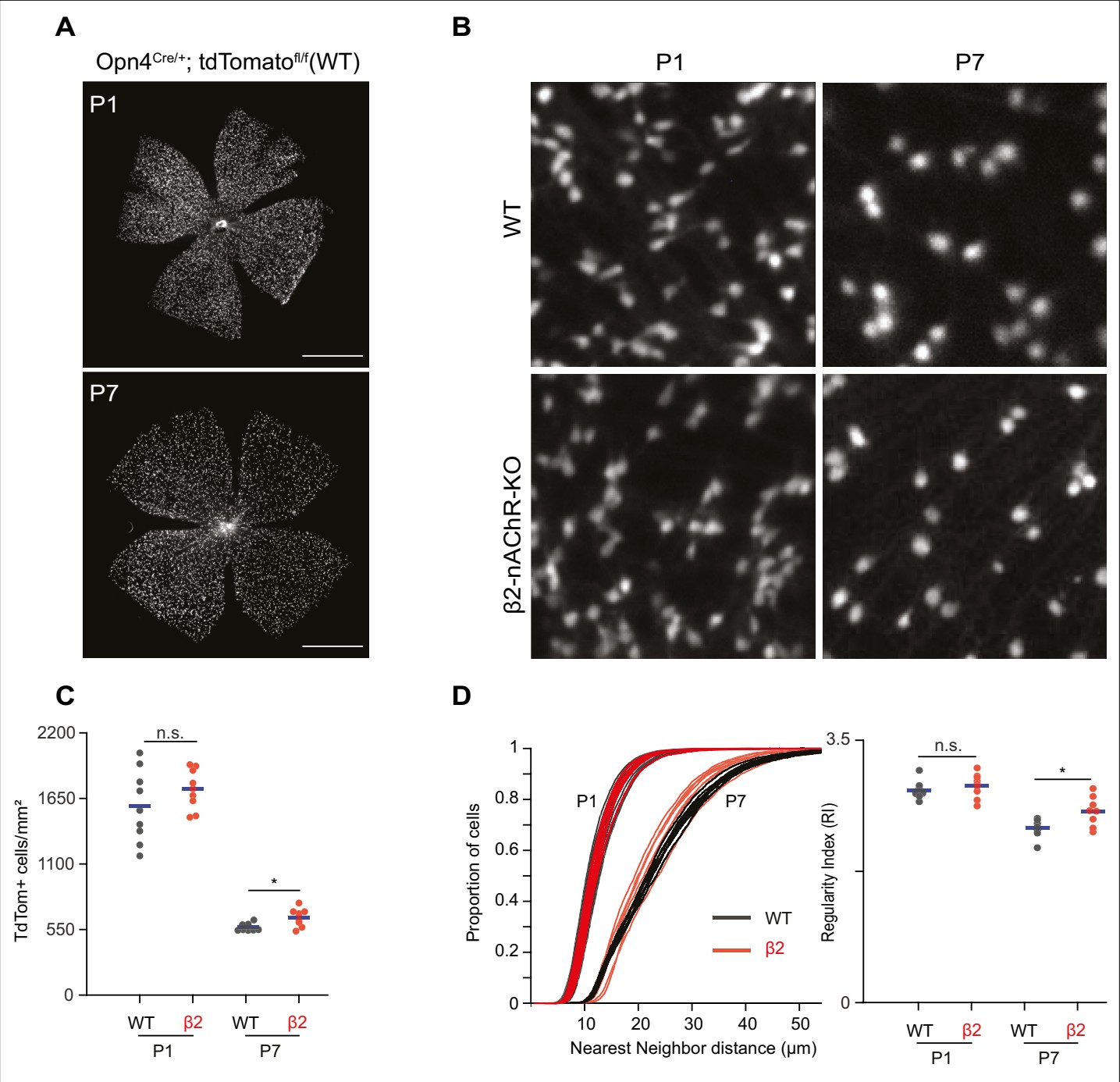

**Figure 5.** Stage 1 and 2 waves do not contribute to the developmental cell death of intrinsically photosensitive ganglion cells (ipRGCs). (**A**) Example epifluorescence images of *Opn4^Cre/+^;Ai9* (WT) retinas at P1 and P7. Scale bars are 500 μm; each field of view is 4.7 × 4.7 mm. (**B**) Representative 200 × 200 μm² fields of view for P1 and P7 WT and ß2-nAChR-KO retina. (**C**) ipRGC densities across different ages and genotypes. P1 WT n = 9 retinas from six mice, P1 ß2-nAChR-KO, 9 retinas, seven mice; P7 WT n = 8 retinas from five mice; P7 ß2-nAChR-KO n = 8 retinas from eight mice. Means represented by blue bars. p=0.012. (**D**) Cumulative distribution function of nearest-neighbor distances (NND) and regularity index (mean NND/SD) from individual retinas, separated by genotype and age. Means represented by blue bars. p=0.020.

The online version of this article includes the following source data for figure 5:

**Source data 1.** Cell density analysis and regularity index.

**Table 2.** Nearest-neighbor distances (NND) and regularity indices for intrinsically photosensitive ganglion cells (ipRGCs) labeled in *Opn4*[Cre/+];*Ai9* mice.
Data represented as averages ± SD. This data complements quantification in *Figure 5*.

|  | Median nearest-neighbor distance (μm) | Regularity index (NND/SD) | Number of retinas |
|---|---|---|---|
| WT P1 | 11.59 ± 0.93 | 2.83 ± 0.12 | 9 |
| β2 P1 | 11.51 ± 0.55 | 2.89 ± 0.16 | 9 |
| WT P7 | 22.32 ± 0.58 | 2.33 ± 0.13 | 8 |
| β2 P7 | 20.96 ± 1.40 | 2.55 ± 0.20 | 8 |

that the proliferation and cell death processes that influence ipRGC density are not dependent on normal patterns of wave activity.

## Distinctions and similarities between stage 1 and stage 2 waves

Our findings support a model in which there are both key differences and some similarities between stage 1 and 2 waves. The distribution of ITI is peaked at roughly 80 s for stage 1 waves that is a slightly longer interval than the peak interval reported for stage 2 (compare *Figure 1C* here to Figure 7D in *Ford et al., 2012*). Stage 1 and 2 waves also propagate at similar speeds. Key differences are that spontaneous activity between E16 and E18 included small non-propagating transients and propagating waves of a broad range of sizes. In contrast, after P1, there are no small non-propagating transients, and stage 2 waves mostly propagate over large areas of the retina, as observed both in vitro (*Feller et al., 1997*; *Hilgen et al., 2017*) and in vivo (*Ackman et al., 2012*). Note the spatiotemporal properties of stage 2 waves also vary dramatically across the developmental period spanning P1–P10 (*Ge et al., 2021*; *Hilgen et al., 2017*; *Maccione et al., 2014*) and therefore we are making these comparisons to the earlier stage 2 waves.

The similarity in frequency of waves and propagation speed indicates that the mechanisms responsible for initiating and propagating large waves are similar for stage 1 and 2 waves. Indeed, stage 1 and 2 waves share a dependence on nAChR signaling. Starburst amacrine cells (SACs) are the sole source of ACh in the retina. During the period of development concurrent with stage 1 waves, SACs begin to migrate toward the INL and send projections to the IPL via homotypic contacts (*Ray et al., 2018*). Also during this period, SACs start to express choline acetyltransferase at E17 in the rat retina (*Kim et al., 2000*), equivalent to E15.5 in mice (*Schneider and Norton, 1979*), and show a response to nicotine in the fetal rabbit retina between E20 and 27 (*Wong, 1995*), corresponding to stage 1 and 2 waves in rabbit (*Syed et al., 2004*). Together these studies support the idea that during embryonic development, SACs are not only releasing ACh but also forming a cholinergic network, similar to that of stage 2 wave propagation (*Ford and Feller, 2012*). Here, we have shown that stage 1 waves fail to initiate in the presence of the general nAChR antagonists, hexamethonium (Hex) and epibatidine (EPB). This suggests that spontaneous depolarization of SACs is important for wave initiation and that the cholinergic circuits that mediates waves start earlier than expected.

One key difference in ACh signaling between stage 1 and 2 waves is their sensitivity to the specific nAChR antagonist, DHβE. Though DHβE reduced the number of stage 1 waves, many smaller calcium transients persisted. In contrast, DHβE is a potent blocker of all activity during stage 2 waves (*Ford et al., 2012*). DHβE is an nAChR antagonist with a greater affinity for nAChRs containing α4 and β2 subunits in heterologous systems (*Ho et al., 2020*; *Papke et al., 2010*). The fact that all stage 1 wave activity is blocked by Hex and EPB suggests that different subunit combinations of nAChRs, on both SACs and RGCs, are independently contributing to either the initiation or propagation of stage 1 waves.

The different sensitivity to nAChR antagonists between stage 1 and 2 waves is highlighted in the patterns of retinal waves in β2-nAChR-KO mice. β2-nAChR-KO mice have significantly reduced stage 2 cholinergic waves (*Bansal et al., 2000*; *Burbridge et al., 2014*; *Xu et al., 2015*; *Xu et al., 2016*) and as such have served as the canonical model for studying the role of stage 2 cholinergic waves in eye-specific segregation, retinotopic maps, retinal and collicular direction selectivity, and in the optokinetic reflex (*Arroyo and Feller, 2016*; *Grubb et al., 2003*; *Thompson et al., 2017*; *Tiriac et al., 2022*; *Wang et al., 2009*). In contrast, stage 1 waves in the β2-nAChR-KO mice

persist, albeit they propagate with slower speed and cover smaller areas of the retina (*Figure 3*). Despite this difference in spatiotemporal properties, both stage 1 and 2 waves that persist in the β2-nAChR-KO mice are blocked by gap junction receptor antagonist rather than blockers of fast neurotransmitter receptors (*Kirkby and Feller, 2013*). Recent evidence suggests that embryonic activity influences many aspects of visual system development including eye specific segregation and retinotopic mapping of retinal projections to the brain (*Guillamón-Vivancos et al., 2022*), as well as cortical innervation of the visual thalamic nucleus (*Moreno-Juan et al., 2023*). Whether the various visual system phenotypes observed in β2-nAChR-KO mice can be attributed in part to reduced stage 1 waves remains to be determined.

In our hands, global blockade or desensitization of nAChRs completely abolished stage 1 waves. This result appears to conflict with previous studies in mice showing that application of curare, a competitive antagonist for nAChRs, preserves small non-propagating transients (*Bansal et al., 2000*). One possibility is that in contrast to hexamethonium, curare has mixed affinity for neuronal nAChRs. A second is that curare was acting via other neurotransmitter receptors where it has some cross-reactivity not shared by the receptor antagonists that we used (*Spirova et al., 2019*; *Wotring and Yoon, 1995*).

In addition to the ACh signaling, gap junctions also play a role in mediating stage 1 retinal waves. Here, we use the gap junction antagonist meclofenamic acid (MFA), which was previously shown to reversibly block junctional conductance (*Veruki and Hartveit, 2009*), dye coupling (*Pan et al., 2007*), and spikelets between developing RGCs (*Caval-Holme et al., 2019*). Application of MFA led to a significant reduction in both the frequency and size of stage 1 waves. The results we observed in MFA are consistent with the pharmacological studies of stage 1 waves in other species: stage 1 waves in rabbit are insensitive to nAChR antagonists (*Syed et al., 2004*), as are early waves in developing chick retina (*Catsicas et al., 1998*). Note that waves in these species were also sensitive to antagonists of various metabotropic receptors, indicating that neurotransmitters are still important for propagating waves in these systems. However, there is the important caveat that MFA can also have some off-target effects that might impact wave propagation (*Kuo et al., 2016*). Previously we showed that in P6-7 retina MFA did not alter depolarization-induced calcium transients and did not reduce the excitability of RGCs. Here, we show that short applications (10 min) of MFA increased input resistance of RGCs and did not appear to block compound EPSCs associated with waves or voltage-gated sodium and potassium channels. Future experiments with independent measures of gap junction coupling – such as tracer coupling and potentially knockout of connexin proteins – are necessary to have a more complete understanding of how electrical synapses contribute to wave propagation properties.

We also observed a difference between stage 1 and 2 waves in β2-nAChR-KO retinas. Embryonic β2-nAChR-KO retinas still exhibited stage 1 waves but at reduced frequency and size when compared to WT retinas. Though the spatiotemporal properties of stage 2 waves recorded in vitro for P1-P8 β2-nAChR-KO retinas are highly dependent on recording conditions with results ranging from sparse activity to high frequency (*Bansal et al., 2000*; *Stafford et al., 2009*; *Xu et al., 2016*), in vivo waves in β2-nAChR-KO are infrequent and weakly depolarizing (*Burbridge et al., 2014*). The activity that persists in β2-nAChR-KO retinas both embryonically and postnatally is resistant to nAChR antagonists, as was observed previously (*Bansal et al., 2000*). Rather the remaining activity in β2-nAChR-KO retinas is completely blocked by MFA, thereby suggesting that stage 1 and 2 waves in β2-nAChR-KO retinas rely solely on gap junctions (*Kirkby and Feller, 2013*).

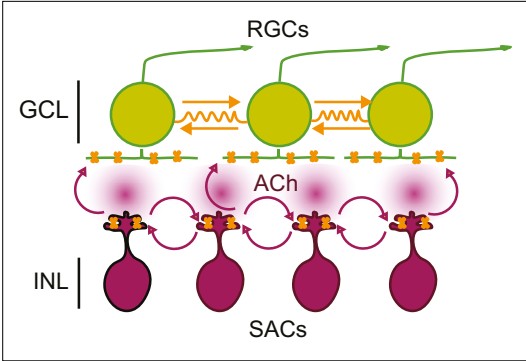

**Figure 6.** Working model of stage 1 wave initiation and propagation. Wave initiation set by spontaneously depolarizing cholinergic amacrine cell/SAC, outlined in black, which release ACh and depolarizes neighboring cells, leading to the volumetric release of ACh. Retinal ganglion cells (RGCs) depolarized by nAChR activation (orange) via SAC-induced ACh release. Wave propagation set by RGC depolarization via gap junctional currents (orange) and volumetric ACh release.

We propose a model for stage 1 waves consistent with these observations (*Figure 6*). Stage 1 wave initiation is dependent on the spontaneous depolarization of SACs, but wave propagation is dependent on both gap junction coupling and nAChR activation. Hence, stage 1 wave initiation is similar to stage 2 waves where the spontaneous depolarization of a SAC activates neighboring SACs, leading to the volumetric release of ACh responsible for wave propagation both in the INL and GCL (*Ford et al., 2012*). The model for stage 1 wave propagation is also similar to stage 2 wave propagation, though a complete description of how gap junction coupling is mediating stage 2 wave propagation is needed. The pharmacological and genetic block/ablation of gap junctions have yet to reveal a phenotype (*Blankenship et al., 2011*; *Caval-Holme et al., 2019*; *Kirkby and Feller, 2013*; *Singer et al., 2001*; *Torborg et al., 2005*). Interestingly, pure gap junction-mediated waves, such as those we observed in the β2-nAChR KO mouse, are considerably slower than waves in WT retina. A recent computational model of gap junction-mediated stage 1 waves, where waves are initiated by RGCs undergoing rare and random depolarizations that propagate entirely via electrical synapses, argues that the speed of propagation is limited by the slow rate at which the junctional currents charge up the membrane capacitance of neighboring RGCs (*Kähne et al., 2019*). Hence, the faster speed of waves mediated by a combination of nAChRs and gap junctions indicates that diffuse release of ACh leads to faster propagation than electrical synapses alone.

## Interactions between stage 1 waves and ipRGCs

Early intrinsic light responses of ipRGCs have been implicated in several developmental events (*Aranda and Schmidt, 2021*), including retinal vascularization (*Rao et al., 2013*), maturation of circadian circuits (*McNeill et al., 2011*), and the maturation of the lens to prevent myopia (*Chakraborty et al., 2022*). Here, we report that ipRGCs are robustly activated by stage 1 waves, similar to our observations that ipRGCs participate in stage 2 waves (*Arroyo et al., 2016*; *Caval-Holme et al., 2022*; *Kirkby and Feller, 2013*). Thus, it is possible that depolarization via stage 1 waves may contribute to some of these ipRGC-dependent developmental processes.

To begin to explore to role of stage 1 and 2 waves in ipRGC development, we monitored the impact of chronically altered waves on the distribution and density of ipRGCs across the retina. Notably, ipRGCs undergo extensive apoptotic cell death, with the peak of apoptosis occurring between P2–P4 (*Chen et al., 2013*). Prevention of apoptosis during this developmental period doubles the density of ipRGCs and dramatically increases the clumping of M1-ipRGC somas. In some systems, correlated network activity has been implicated in cell proliferation and cell death. For example, retinal wave activity promotes neurite outgrowth and potentially survival among RGCs (*Goldberg et al., 2002*). Additionally, in the developing primary somatosensory and motor cortices, higher levels of spontaneous electrical activity were shown to have a neuroprotective effect (*Blanquie et al., 2017b*). However, here we show in the β2-nAChR-KO mouse, which has significantly diminished stage 1 and 2 retinal waves, the normal developmental loss of ipRGCs between P1 and P7 is maintained. Thus, stage 1 and 2 retinal waves are not required for ipRGC apoptosis; however, it is possible that the residual wave activity in β2-nAChR-KO is sufficient to activate pro-survival pathways. A deeper understanding of how spontaneous activity modulates RGC survival pathways (e.g., *Ahmed et al., 2022*) is warranted.

## Methods

### Animals

All animal procedures were approved by the UC Berkeley Institutional Animal Care and Use Committee and conformed to the NIH Guide for the Care and Use of Laboratory Animals (AUP-2015-10-8080-2), the Public Health Service Policy, and the SFN Policy on the Use of Animals in Neuroscience Research. For our calcium dye-based calcium imaging, *Opn4$^{Cre/+}$;Ai9* mice were generated by crossing mice *Opn4$^{Cre/+}$* B6.Cg-*Gt(ROSA)26Sor$^{tm9(CAG-tdTomato)\ Hze}$*/J mice (Ai9) mice (stock # 007909, Jackson Laboratory, Bar Harbor, ME) to the *Opn4$^{Cre/+}$* reporter mouse (T. Schmidt, Northwestern University, Evanston, IL). For our GCaMP6s-based calcium imaging, we generated *Vglut2;GCaMP6s* mice by crossing B6J.129S6(FVB)-*Slc17a6$^{tm2(cre)Lowl}$*/MwarJ mice (stock # 028866) to B6J.Cg-*Gt (ROSA)26Sor$^{tm96(CAG-GCaMP6s)Hze}$*/MwarJ mice (stock # 028863). ipRGC density measurements were conducted on P1–P7 mice of either sex using *Opn4$^{Cre/+}$;Ai9*.

To obtain mice that were precisely at the correct embryonic age, we set up timed pregnancies and checked vaginal plugs every morning for 4 days after the animals were paired. This approach led to an uncertainty of age of ± 1 day. Hence, we grouped data across E16–18. Since we observed more variance within litters than across litters, we assume age was not a determining factor in our findings. We used the β2-nAChR-KO mouse line in which the β subunit of the nicotinic acetylcholine receptor is knocked out as a genetic model in which cholinergic retinal spontaneous activity is disrupted. For experiments regarding the influence of spontaneous retinal activity on the distribution of ipRGCs across the retina, we used β2$^{-/-}$::$Opn4^{Cre/+}$;$Ai9$ (β2-nAChR-KO) mice, generated by crossing β2$^{-/-}$ (A. Beaudet, Baylor University, Waco, TX) mice to $Opn4^{Cre/+}$;$Ai9$ mice to label all melanopsin-expressing cells. All mouse lines are maintained on a C57BL/6 genetic background. All animals used for two-photon calcium imaging experiments and immunohistochemistry were housed in 12 hr day/night cycle rooms.

## Retinal preparation

On the day of the experiment, pregnant dams were deeply anesthetized via isoflurane inhalation and fetuses were harvested via a cesarean section. tdTomato-positive fetuses were identified using miner goggles (Biological Laboratory Equipment Services and Maintenance Ltd., model: GFsP-5). Fetuses were kept alive in 50 ml Falcon tubes filled with oxygenated (95% $O_2$ 5% $CO_2$) ACSF (in mM, 119 NaCl, 2.5 KCl, 1.3 $MgCl_2$, 1 $K_2HPO_4$, 26.2 $NaHCO_3$, 11 D-glucose, and 2.5 $CaCl_2$). Fetuses were then euthanized sequentially by decapitation. Eyes were immediately enucleated and retinas were dissected at room temperature in oxygenated ACSF under a dissecting microscope. Isolated retinas were mounted whole over a 1–2 mm$^2$ hole in nitrocellulose filter paper (Millipore) with the photoreceptor layer side down and transferred to the recording chamber of a two-photon microscope for imaging. The whole-mount retinas were continuously perfused (3 ml/min) with oxygenated ACSF warmed to 32–34°C by a regulated inline heater (TC-344B, Warner Instruments) for the duration of the experiment. Additional retina pieces were kept in the dark at room temperature in ACSF bubbled with 95% O2, 5%CO2 until use (maximum 8 hr).

For the calcium imaging experiments, retinas were bath loaded with the calcium indicator Cal 520 AM (AAT Bioquest) for 1–2 hr at 32°C.

## Two-photon calcium imaging

Two-photon fluorescence measurements were obtained with a modified movable objective microscope (MOM) (Sutter instruments, Novator, CA) and made using an Olympus ×60, 1.00 NA, LUMPlanFLN objective (Olympus America, Melville, NY) for single-cell resolution imaging (field of view, FOV: 203 × 203 µm) or a Nikon ×16, 0.80 NA, N16XLWD-PF objective (Nikon, Tokyo, Japan) for large FOV (850 × 850 um) imaging. Two-photon excitation was evoked with an ultrafast pulsed laser (Chameleon Ultra II; Coherent) tuned to 920 nm to image Cal520, GCaMP6s, and tdTomato. Laser power was set between 6.5 and 12 mW for imaging of Cal520 and tdTomato expression. The microscope system was controlled by the ScanImage software (https://www.scanimage.org/). Scan parameters were [pixels/line × lines/frame (frame rate in Hz)]: [256 × 256 (1.48 Hz)], at 2 ms/line. This MOM was equipped with a through-the-objective light stimulation and two detection channels for fluorescence imaging.

## Epifluorescent macroscope calcium imaging

Epifluorescent calcium imaging were obtained on a custom-built macroscope with an Olympus XLFLUOR4X/340 4× 0.28 NA objective, a Teledyne Kinetix camera. Collectively, this macroscope has 4.7 mm × 4.7 mm FOV, and 1.5 µm/pixel. All movies were taken at a 10 Hz frame rate and pixels were binned 4 × 4 bringing the resolution down to 5.9 µm/pixel, still maintaining single-cell resolution. Cal520 and GCaMP6s excitation was evoked with a 474 nm LED. A full description and building instructions can be found at https://github.com/Llamero/DIY_Epifluorescence_Macroscope, (*Smith, 2022*).

## Initiation site measurements

Macroscope recordings of stage 1 events were used for the manual detection of initiation sites. Small non-propagating events were identified as local regions of correlated calcium activity with fixed areas and no wave fronts. For this analysis only, we separated waves into small and large events to see

whether there were differences in initiation sites. Small propagating waves were identified as regions of correlated calcium activity with no fixed areas and with wave fronts covering up to 25% of the retina. Large propagating waves were identified as regions of correlated calcium activity with no fixed areas and with wave fronts covering up to 90% of the retina.

## Pharmacology

We blocked gap junctions via application of the gap junction blocker MFA (50 µM, Sigma-Aldrich). We blocked the nicotinic acetylcholine signaling pathway via application of the broad nicotinic receptor antagonists hexamethonium (Hex, 100 µM, Sigma-Aldrich) and epibatidine (EPB, 10 nM, Sigma-Aldrich) as well as the specific antagonist dihydro-ß-erythroidine hydrobromide (DHßE, 8 µM, Sigma-Aldrich).

The following procedure was used for all pharmacology experiments: We recorded baseline activity in ACSF for 8 min before pharmacological agents were applied to the perfusion system. We then waited 15–30 min for the agents to take effect before acquiring another 8 min recording session.

To attempt to assay off-target effects of MFA, we used whole-cell voltage-clamp recordings to compare voltage-gated ion channels on RGCs in E16–18 retina but found inconsistent results (*Figure 2—figure supplement 1*). We associate this high variance with a rapid changing complement of ion channels during development and the quick washout of these conductances during whole-cell recordings.

## Image analysis of population calcium imaging movies

Movies were preprocessed for motion correction using a MATLAB code from the Flat Iron Institute (https://github.com/flatironinstitute/NoRMCorre; *Pnevmatikakis, 2019*). The baseline movie frame (F0) was computed by taking the temporal median projection of all the movie frames. Each movie frame (F) was normalized by dividing its difference from the baseline frame (F-F0) by the baseline frame ((F-F0)/F0) to produce a ΔF/F0 movie. For movies taken on the two-photon microscope, circular ROIs were drawn on all cells within the FOV. Additional circular ROIs were drawn for tdTomato+ cells. For movies taken on the macroscope a grid of 10 µm × 10 µm squares, which were spaced 1.5 pixels apart, were drawn over the whole surface of the retina using a custom FIJI macro. The ROIs and the ΔF/F0 movie were then imported into MATLAB for further analysis using custom algorithms. Traces for each FOV and ROI were computed as the mean value of the pixels enclosed by the ROI in each frame of the ΔF/F0 movie.

For transient frequency analysis, event detection was done using the findpeaks function in MATLAB, with the minimum threshold set to greater than at least 10 times the standard deviation of the baseline fluorescence, which corresponds to at least 10% $\Delta F/F_0$. The ITI for each ROI was calculated by finding the difference between the frame of each detected transient. This difference was then converted to seconds by multiplying it to the movie's frame rate.

The area of waves was calculated for all movies taken on the macroscope. To do this, we first z-scored the percent active ROI traces and used the findpeaks function to detect individual waves, with the threshold set to a z-score of 1. After determining the time of waves, we removed any waves that occurred within the first and last 6 s of the movie due to them being edge cases. We then summed the number of active ROIs within 12 s around the wave times. This number was then divided by the total number of ROIs to get the percentage of active ROIs.

To determine whether neurons participate in waves in the two-photon calcium imaging data, we employed the following bootstrapping strategy: we randomly sampled the activity of individual neurons outside of wave times a thousand times to build a baseline of neural activity. We then set a threshold of 95th percentile to statistically determine whether neurons exhibited a greater calcium response during a wave than at rest. We then calculated the percentage of waves each cell participated in and averaged this value for every FOV. Similarly, mean response amplitudes were calculated for each cell and then averaged for each FOV.

For a description of statistics used, please refer to the figure captions.

## Analysis of ipRGC densities

To image the density of ipRGCs in fixed retinas, dissected retinas from P1 and P7 mice were fixed in 4% PFA for 30 min. The fixed retinas were subsequently mounted on a slide with vectashield and a

cover slip, then imaged within an hour of mounting on the macroscope. For P1 retinas, Z-stacks were acquired by manually turning the focus knob.

We first identified the centroid of each ipRGC. For P7 retinas, where there is more space between cells, we employed the following automatic segmentation. Images were bandpass filtered and somata automatically segmented using the Morpholibj (*Legland et al., 2016*) classic watershed tool to obtain 8-bit binarized masks. The masks were then processed in MATLAB in order to obtain the centroid locations and nearest-neighbor distances for each soma. For P1 retinas, where there is less space between cells and in fact cells seem to form clusters, automatic segmentation was not possible. Therefore, cells were manually marked using the ImageJ multipoint tool and soma locations exported as a CSV file. For all ages, the centroid data was imported to MATLAB for further analyses.

Density was quantified by dividing the microscope field of view up into 200 ×200 μm squares, manually excluding ones that did not cover the retina or covered partial or damaged parts of the retinas. Out of the resulting squares (~150 per P7 retina, ~100 per P1 retinas), 75 squares (for P7 retinas) and 50 squares (for P1 retinas) were randomly selected and the average density of TdTom+ cells in those squares calculated.

To quantify the nearest-neighbor distances, we used a custom-written MATLAB code that, for each ipRGCs, identified the closest neighbor using the shortest Euclidean distance.

## Statistical tests

Details of statistical tests, number of replicates, and p values are indicated in the figures and figure captions. p values < 0.05 were considered significant.

## Acknowledgements

All authors supported by NIH RO1EY013528, RO1EY019498, and P30EY003176. AT was supported by K99EY030909. BES was supported by NIH P30EY003176. We thank members of the Feller Lab for their comments on the manuscript.

## Additional information

### Competing interests

Marla B Feller: Reviewing editor, eLife. The other authors declare that no competing interests exist.

### Funding

| Funder | Grant reference number | Author |
|---|---|---|
| National Eye Institute | RO1EY013528 | Marla B Feller |
| National Eye Institute | RO1EY019498 | Marla B Feller |
| National Eye Institute | P30EY003176 | Marla B Feller Benjamin E Smith |
| National Eye Institute | K99EY030909 | Alexandre Tiriac |

The funders had no role in study design, data collection and interpretation, or the decision to submit the work for publication.

### Author contributions

Christiane Voufo, Conceptualization, Software, Formal analysis, Validation, Investigation, Visualization, Methodology, Writing – original draft, Writing – review and editing; Andy Quaen Chen, Conceptualization, Software, Formal analysis, Investigation, Visualization, Methodology, Writing – original draft, Writing – review and editing; Benjamin E Smith, Software, Methodology, Writing – review and editing; Rongshan Yan, Formal analysis, Investigation; Marla B Feller, Conceptualization, Supervision, Funding acquisition, Writing – original draft, Project administration, Writing – review and editing; Alexandre Tiriac, Software, Supervision, Investigation, Writing – original draft, Project administration, Writing – review and editing

## Author ORCIDs

Christiane Voufo (iD) http://orcid.org/0000-0001-5299-0913
Andy Quaen Chen (iD) http://orcid.org/0000-0002-4970-045X
Marla B Feller (iD) http://orcid.org/0000-0002-9137-5849
Alexandre Tiriac (iD) http://orcid.org/0000-0002-7966-981X

## Ethics

All animal procedures were approved by the UC Berkeley Institutional Animal Care and Use Committee and conformed to the NIH Guide for the Care and Use of Laboratory Animals, the Public Health Service Policy, and the SFN Policy on the Use of Animals in Neuroscience Research.

## Decision letter and Author response

Decision letter https://doi.org/10.7554/eLife.81983.sa1
Author response https://doi.org/10.7554/eLife.81983.sa2

## Additional files

### Supplementary files

• MDAR checklist

### Data availability

We have uploaded the raw data for Figure 5 on Dryad. All other raw imaging data and images are available upon request as they are too large to upload to Dryad. They are residing on our lab server and can be transferred via ftp. Figures 1-4: These data are based on movies acquired from live imaging of activity using a macroscope or a 2-photon scanning microscope. Figure 1: 50 gigabytes Figures 2/4: 180 gigabytes Figure 3: 233 gigabytes Figure 5: These are high resolution fluorescence images acquired from microscope at various z-plane focus planes. 7 gigabytes total (on Dryad). All code and software are available on GitHub: https://github.com/FellerLabCodeShare/Embryonic-Retinal-Waves (copy archived at swh:1:rev:b205760d05ea3c0dcd6aca8ed3f7bcb273f4c5a5). Instructions on how to build a macroscope available at this GitHub repo: https://github.com/Llamero/DIY_Epifluorescence_Macroscope.

The following dataset was generated:

| Author(s) | Year | Dataset title | Dataset URL | Database and Identifier |
|---|---|---|---|---|
| Voufo C, Chen AQ, Smith B, Yan R, Feller MB, Tiriac A | 2023 | Circuit mechanisms underlying embryonic retinal waves | https://doi.org/10.5061/dryad.h18931zr2 | Dryad Digital Repository, 10.5061/dryad.h18931zr2 |

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
