## [Editor Report]

This paper investigates waves in embryonic mouse retinas. These stage 1 waves have been studied less than the post-natal (stage 2) waves. The paper uses elegant imaging and analysis approaches to monitor calcium signals across the entire retina and to determine the properties of the waves and their dependence on cholinergic and electrical synapses. This contributes substantially to the understanding of how these waves are generated.

---

## [Decision Letter]

**Decision letter after peer review:**

Thank you for submitting your article "Cellular Mechanisms Underlying Embryonic Retinal Waves" for consideration by *eLife*. Your article has been reviewed by 3 peer reviewers, including Fred Rieke as Reviewing Editor and Reviewer #1, and the evaluation has been overseen by Marianne Bronner as the Senior Editor. The following individual involved in the review of your submission has agreed to reveal their identity: Guillermina López-Bendito (Reviewer #3).

The reviewers have discussed their reviews with one another, and the Reviewing Editor has drafted this to help you prepare a revised submission. The reviewers agreed that the topic of the paper was important and appreciated the technical advances that made the work possible. Several important issues came up in review, however, that need to be strengthened before considering the paper further.

(1) Possible off-target effects of gap junction inhibitors should be discussed earlier and in more detail.

(2) The characterization of ipRGC development needs to be better motivated.

(3) Different subtypes of activity during the time period when stage 1 waves are observed need to be defined and distinguished more clearly.

(4) Several data analysis issues need attention.

These and a number of other issues are described in detail in the reviews.

*Reviewer #1 (Recommendations for the authors):*

Page 7: The description of Stage 1 waves in rabbits talks about "wave frequency" being insensitive to block of fast synaptic transmission, and "blocked" by gap-junction antagonists. It is not clear whether this means all waves, or if "frequency" is included to indicate a subset of waves or properties. Please clarify!

Page 7, bottom: "Waves that persisted … were smaller and had comparable amplitudes …" – clarify that this means smaller in spatial extent.

Page 8, first sentence: revise – not clear how signaling is comprised of subunits.

Page 8, last full paragraph: this paragraph starts by saying that there is a light-dependent modulation of stage 1 wave frequency, but ends by saying that there is not. Is this a difference between WT and KO? If so that could be clarified.

Page 9, bottom, and page 10 top: Sentence starting "Interestingly …" is confusing. It seems to say that the regularity index is regular, but matches expectations if it was random. Please revise.

*Reviewer #2 (Recommendations for the authors):*

The Introduction is too long and reads more like a review. I suggest limiting it to specific premise, tested hypothesis, and eliminating paragraphs 2 and 3.

Few technical notes:

In the final Results paragraph the cell body size of 7-10 μm is used to estimate ipRGC mosaics. This is too small. Please consider 20-30um.

Across the text, some statistical values are missing.

In Discussion, please note that while MFA does block GJs, it can also reduce spiking activity by a GJ-independent mechanism.

*Reviewer #3 (Recommendations for the authors):*

I provide here a long list of points that the authors should address. Despite the length of the list, the points I raised and requested do not involve any fundamental error or flaw in the manuscript but are aimed to improve the analysis, organization, and presentation of the dataset.

– It has been shown previously that stage 2 waves span from P1-P10. In this manuscript, the authors nicely set the focus on the embryonic phase of stage 1. The properties of both wave patterns are similar in many aspects (propagation speed, IEIs, and to some extent cholinergic). Thus, it is possible to think that the broad features extracted by the authors from stage 1 retinal waves represent the slow maturation of what we know as stage 2 waves rather than a preceding and complete separate class of retinal waves. Could this be the case? Could the authors speculate in the discussion on what happens in the transition between stages 1 and 2, at P0-P1?

– The authors pool together data from E16, E17, and E18 retinas. Are there any differences in the properties of the waves across time? The authors should at least show this data separately in the supplementary materials to justify pooling the data.

– It seems like the authors recognize the gap-junction component to a lesser extent and stress the cholinergic component. As one of the goals of the paper is to characterize the so-far-unknown stage 1 retinal waves, considered to be mainly gap-junction mediated in the literature, the reader expects to see a larger discussion on this issue and a comparison to the published literature.

– The first part of the Results could be better organized. It is not clear to me if the authors try to explain the relevant methods of analysis, illustrate the different types of activity they found (showing individual examples), or describe activity showing population data. In the second paragraph, the authors describe how they acquired the data, this is fine. In the third paragraph, the authors explain how they analyzed the movies and some properties of the transients detected per ROI and how ROIs participate in waves (although there is not, beforehand, a clear definition of what a wave is). Here, some clarity is needed when the authors use the terms "transients", "events", and "waves". Up to this point, the authors refer to figure 1 where they show data from sample experiments. But, in the last paragraph of this section, they move to population data (and jump suddenly to panels of figure 3). Moreover, they focused on large propagating events, though it seems that sometimes they refer to all the propagating events, and present a summary in Table 1, where surprisingly small waves appeared for the first time without being mentioned in the text. In sum, it would be helpful for the reader if the authors reorganize the text of this section, try to set a clear scope for it, and present the data as smoothly and clearly as possible.

– Are waves detected by looking at the "% Active" plots, by visual inspection of the movies or by an algorithm? In the first case, what happens if there are two or more waves running simultaneously? It would help the reader if the authors explain how waves were detected. If wave detection is manual or visual, why the authors did not use automatic detection? In figure S2, it seems that there are a lot of waves per experiment, too many to be quantified manually or by eye.

– I suggest modifying the title to better fit the data. In the first half of the manuscript, the authors investigate the spatiotemporal properties and pharmacological profile of embryonic retinal waves. Using this information, the authors propose a model for the initiation and propagation of stage 1 waves (figure 6), but none of the cell types of the model were directly assessed in the manuscript. Regarding the ipRGCs, the authors nicely show that these cells participate in the network activity at early developmental stages; however, the data does not show the mechanism related to ipRGCs. Strictly, and in my opinion, the cellular mechanisms are not really established here.

– Why is the frequency of waves in the Hex control (Figure 2B) half the frequency of the other controls?

– The authors should include the mean duration of the waves/events in the description of the different types of activity.

– The authors claim that stage 1 retinal waves persist in WT retinas treated with DHβE. It is true that some activity remains upon DhβE, but, to me, the raster plot looks different than the WT (figure 2D). The statistical test yields no significant differences, but the p-value is amazingly borderline p = 0.05. This means that more experiments are needed to raise robust conclusions. Moreover, since DhβE (figure 2E) reduce the area of the events from 90% to 10% … is it still possible to call them stage 1 retinal waves? In sum, when the authors claim that waves persist, it suggests that normal stage 1 waves were observed in the DhβE-treated retinas; however, what remains looks like something different.

– As stage 1 waves are also affected in β2-nAChR KO mice, which is the "standard" model that has been widely used to assess the function of stage 2 waves, this is a good opportunity for the authors to discuss to what extent the conclusions raised using this mouse in the previous literature needs to be re-interpreted.

Additional comments (Introduction is page 1)

– The term "events" is used both to indicate the activation of an ROI and the correlated activation of multiple ROIs (as in waves). It would be easier for the reader if the authors differentiate them. Examples of the first case can be found in the third paragraph of the results: (i) "… traces for each ROI were rasterized based on an event detection algorithm …" or (ii) "We first computed the time between spontaneous events by measuring the inter-event interval (IEI) for each ROI and …". Examples of the second case can be found in the fourth paragraph: (i) "In addition to propagating events …" or (ii) "These events were like those described previously in (Bansal et al., 2000)". In addition, this disambiguation will clarify what they quantify in the "Events/min" plots. In sum, the authors should somehow clearly differentiate the activation of single ROIs from population activity. Perhaps, "transients" is a good option for the activation of single ROIs. It is not clear from the context which class of events is referred to by "These".

– Why are waves divided into small and large events? This subdivision is not used in any of the quantifications that follow figure 1. In the same line, the authors adequately classified the events in propagating and non-propagating but we don`t know what happens with the latter in the different experimental conditions analyzed. It would be interesting to know whether the properties of non-propagating events are changed or remain the same upon pharmacological or genetic manipulations.

I think that the description of how to classify large waves is not accurate. In Methods: "Large propagating waves were identified as regions of correlated calcium activity with no fixed areas and with wavefronts covering up to 90% of the retina". I guess that the authors meant wavefronts covering areas larger than 25% of the retina, which is the upper limit for small waves. In addition, the authors classify the events as fixed or non-fixed. However, it is not clear to the reader what "fixed" means.

– The number of asterisks used to indicate statistical significance is not consistent (see Figures 2 and 3, for example).

– The algorithm that detects calcium transients within ROIs should be described or referenced.

– Why does the threshold for detection vary from 15% to 50% depending on the experiment? In addition, these thresholds seem very high. Why not use n-times as the SD of the baseline? Is it related to the signal-to-noise ratio?

– I have some comments that may help to refine the Abstract. In lines 2-3, the authors should mention the species. In line 5, as far as I know, stage 1 waves are also observed at least at P0-P1, it is thus not completely correct to use "embryonic retinal waves" as a synonym for stage 1 waves. Perhaps "the earliest pattern of retinal waves" will do it. In line 7, it is not clear to me what the authors mean by "finite". And finally, by reading only the Abstract, it is not evident which could be the relationship between stage 1 waves and the distribution of ipRGCs. Perhaps, the authors should briefly mention here what normally happens to this subpopulation during perinatal development.

– Page 1, line 7. Typo: vision instead of "vison". Page 2, line 26. Typo: characterized instead of "characterize".

– Page 2, lines 29-30. The details of FOV and resolution should go to Methods. The authors emphasize here that the field of view could cover the whole retina while preserving cellular (not subcellular) resolution, this is clear even without the details, increasing the readability of the paragraph. Please, check that "FOV" is spelled out before being abbreviated in the main text.

– Do the authors think that "complex" is a good descriptor for the spatiotemporal properties of stage 1 waves (as stated in the first subheading of Results)?

– I think that the initial paragraph of the first part of Results, about the cell types in the developing retina, does not have a clear link to the following paragraphs of the section. Perhaps, this information fits better in the Introduction. Further, I believe that Bansal and colleagues is not a convenient reference for the last sentence where gap junctions are proposed to be a primary substrate of stage 1 waves. There is only one mention of gap junctions in this publication. The authors may add here the Wong and colleagues (1998).

– Page 3, line 5. I think that it is more appropriate to give the average area of these events in square microns rather than in the number of ROIs occupied by them.

– Page 3, line 23. "E16-18 mouse pups". Better replace mouse pups with "mice" or "mouse embryos".

– Page 4, line 11. Figure 3D is not the correct panel for these data.

– Page 5, line 22. "We observed a dramatic decrease in the density of ipRGCs in WT retinas from P1 to P7 (Figures 5B, C), …". Although differences are evident, this statement needs statistical analysis.

– The authors claim that IEIs are longer in beta2-nAChR-KO than in WT retinas (figure 3E). A statistical analysis should be provided here that directly compares these two populations (Black and pink violins in the panel).

– In the caption of figure 1B, I guess that "an E17" means "an E17 retina".

– In the caption of figure 2, p-values do not correspond to the panels.

– It is convenient that the authors add the maximum number of ROIs represented to the X-axis of raster plots (for instance, 1 at the bottom and say 500 at the top of the axis).

– In Figures 2F and H, the authors show time-series data points; however, there are points without lines in the graphs.

– Rows in Table 1 are individual experiments. Please, add this information to the Table.

– In the speed quantification, the authors explain that the first point of the wavefront is selected at random. I understand then that the speed value for a wave will change every time that the quantification is repeated (because wavefronts do not seem to propagate homogeneously). Is this correct? Or, are measurements repeated many times for every wave and a mean calculated? Also, when the second point is selected, it is not clear to me what "parallel to the direction of the first point" means. This section will benefit from a better explanation of the quantification method. Is this method standard or comparable to other methods that have been used in previous publications?

– Could the authors state in each figure the number of independent experimental units (N): how many retinas from how many mice (for example, information is incomplete in figure 3 and missing in figure 5)?

– "Cook and Bekker, Physiology, 2009" is not listed in the reference list. "Emanuel and Do, 2015" is not listed in the reference list.

[Editors’ note: further revisions were suggested prior to acceptance, as described below.]

Thank you for resubmitting your work entitled **"**Circuit Mechanisms Underlying Embryonic Retinal Waves**"** for further consideration by *eLife*. Your revised article has been evaluated by Marianne Bronner (Senior Editor) and Fred Rieke (Reviewing Editor).

The paper has improved in revision, and most of the original issues have been nicely handled. A few issues remain or have been introduced in the revisions.

MFA. The additional discussion of MFA is helpful though, as noted in the paper, ultimately not entirely conclusive. Did you try to wash out the effects of MFA in any of the experiments described here? Even a partial washout would be helpful. I would consider moving the whole cell recordings on lines 155-158 to the Methods since the variability limited the conclusions that can be drawn from them. It does look like K channel activity decreased systematically in MFA, which might be noted.

The motivation for the measurements in the knockout animals is clearer now. The conclusion that compensation causes gap junctions to contribute more strongly to waves in the knockout mice could be highlighted more (it currently comes in the middle of the paragraph starting on line 180). I would consider adding that conclusion to the abstract or introduction and making it clearer in the results.

The abstract is quite long and detailed. I would suggest removing some of the sentences about the experimental approach to the final paragraph of the introduction, and focusing the abstract more on conclusions to make it more generally and easily accessible.

ipRGC development and waves. Since stage 1 waves are modified but not eliminated in the knockouts, some of the conclusions about waves and ipRGC development seem too strong (e.g. lines 250-251, line 265, lines 283-284). The last sentence of the Discussion strikes me as a balanced way to handle this issue, so some of that could come into the earlier statements.

---

## [Author Response]

The reviewers have discussed their reviews with one another, and the Reviewing Editor has drafted this to help you prepare a revised submission. The reviewers agreed that the topic of the paper was important and appreciated the technical advances that made the work possible. Several important issues came up in review, however, that need to be strengthened before considering the paper further.(1) Possible off-target effects of gap junction inhibitors should be discussed earlier and in more detail.

We have added additional experiments based on whole cells recordings to address some off target effects of MFA but we do make note of the limitations of these new controls since we observed significant variability of voltage-gated conductances across RGCs at this age as well as the limited ability to maintain stable recordings for the requisite time to have within cell controls for MFA. In the revised manuscript we are more explicit about the caveats of using MFA both in results and in discussion.

(2) The characterization of ipRGC development needs to be better motivated.

We felt this was a good cell type with which to start since the reduction of ipRGC due to cell death is well characterized (see Chen et al., Neuron, 2013) while most other studies generalize across all RGCs. We have clarified this motivation in the revised manuscript.

(3) Different subtypes of activity during the time period when stage 1 waves are observed need to be defined and distinguished more clearly.

In the revisions, we tried to develop our analysis so that we could statistically differentiate between small and large waves. Upon further analysis, we found that wave sizes cover a broad range that is not bimodal. Thus, in this revision, we only distinguish between propagating waves and small non-propagating events (as described in revised text, bottom of page 4).

(4) Several data analysis issues need attention.

We hope to have addressed all of these in the revised text.

Reviewer #1 (Recommendations for the authors):Page 7: The description of Stage 1 waves in rabbits talks about "wave frequency" being insensitive to block of fast synaptic transmission, and "blocked" by gap-junction antagonists. It is not clear whether this means all waves, or if "frequency" is included to indicate a subset of waves or properties. Please clarify!

We have clarified this terminology on page 5. We reproduce sentences here for the benefit of reviewer.

“Previous work based on epifluorescent calcium imaging experiments performed in embryonic mice has shown that retinal waves, as defined by correlated changes in fluorescence, are reduced in frequency and size by curare, a competitive antagonist for nAChRs (Bansal, 2000). However, in roughly the equivalent developmental period in rabbit, blockade of all fast neurotransmitter receptors, including nAChRs, had no impact on wave frequency (Syed et al., 2004). Rather, waves in rabbit are blocked after the application of 18β-glycyrrhetinic acid, a gap junction antagonist (Syed et al., 2004).”

Page 7, bottom: "Waves that persisted … were smaller and had comparable amplitudes …" – clarify that this means smaller in spatial extent.

We have revised text to read (page 6)

“However, in contrast to Stage 2 wave, some waves persisted in the presence of DhβE, but they recruited fewer neurons and therefore had smaller areas (Figure 2B).”

Page 8, first sentence: revise – not clear how signaling is comprised of subunits.

We have revised the text to read (page 6):

“Our results thus far indicate that both the frequency and area of Stage 1 retinal waves are modulated by the activation of different subtypes of nAChRs as well as gap junction coupling.”

Page 8, last full paragraph: this paragraph starts by saying that there is a light-dependent modulation of stage 1 wave frequency, but ends by saying that there is not. Is this a difference between WT and KO? If so that could be clarified.

We previously reported that spontaneous correlated activity in the β2-nAChR KO during the Stage 2 waves was light sensitive while spontaneous correlated activity in WT mice is not (Kirkby et al., 2013). We found this not to be the case in β2-nAChR KO mice Stage 1 waves. We hope the revised paragraph on page 6 clarifies this point.

Page 9, bottom, and page 10 top: Sentence starting "Interestingly …" is confusing. It seems to say that the regularity index is regular, but matches expectations if it was random. Please revise.

We thank the reviewer for point out this error. We have revised the text to say that the regularity indices we measured are consistent with a random distribution.

Reviewer #2 (Recommendations for the authors):The Introduction is too long and reads more like a review. I suggest limiting it to specific premise, tested hypothesis, and eliminating paragraphs 2 and 3.

We have streamlined the intro. We do not completely eliminate paragraphs 2 and 3 since they provide background on circuits that mediate waves which we thought might be important for context. However, we did trim greater than 30% of the words.

Few technical notes:In the final Results paragraph the cell body size of 7-10 μm is used to estimate ipRGC mosaics. This is too small. Please consider 20-30um.

We have measured the somas of ipRGCs at these ages and 10 µm is the best estimate value. It is worth noting that given the analysis in Keely et al., larger somas would lead to an increase in the regulatory index associated with a random distribution. Hence, our conclusion that the organization of ipRGCs we observed in WT and β2-nAChR retinas do not vary from random remains valid.

Across the text, some statistical values are missing.

We have added p-values to all Figure captions.

In Discussion, please note that while MFA does block GJs, it can also reduce spiking activity by a GJ-independent mechanism.

We have added discussion regarding off-target effects of MFA earlier in manuscript in presentation of the results, as suggested by Reviewer #1. Please see pages 5 and 10 in revised manuscript.

Reviewer #3 (Recommendations for the authors):I provide here a long list of points that the authors should address. Despite the length of the list, the points I raised and requested do not involve any fundamental error or flaw in the manuscript but are aimed to improve the analysis, organization, and presentation of the dataset.– It has been shown previously that stage 2 waves span from P1-P10. In this manuscript, the authors nicely set the focus on the embryonic phase of stage 1. The properties of both wave patterns are similar in many aspects (propagation speed, IEIs, and to some extent cholinergic). Thus, it is possible to think that the broad features extracted by the authors from stage 1 retinal waves represent the slow maturation of what we know as stage 2 waves rather than a preceding and complete separate class of retinal waves. Could this be the case? Could the authors speculate in the discussion on what happens in the transition between stages 1 and 2, at P0-P1?

We revised the discussion significantly based on these comments. Please see pages 9 -11.

– The authors pool together data from E16, E17, and E18 retinas. Are there any differences in the properties of the waves across time? The authors should at least show this data separately in the supplementary materials to justify pooling the data.

We do not feel confident that we could rely on plug checks to estimate age of embryos better than +/- one day. Hence we have grouped across ages. We have assessed whether any variance could be attributed to mice that came from different litters (as a proxy for potential different ages). However we found more variance within litters than across litters and therefore we assume age does not explain the variance. We have added this to the Methods-Animals section.

– It seems like the authors recognize the gap-junction component to a lesser extent and stress the cholinergic component. As one of the goals of the paper is to characterize the so-far-unknown stage 1 retinal waves, considered to be mainly gap-junction mediated in the literature, the reader expects to see a larger discussion on this issue and a comparison to the published literature.

The dependence on nAChR signaling was more surprising to us given the lack of synaptic structures. In the revised manuscript, we think we have made clearer that both gap junction coupling and nAChR signaling are required for Stage 1 waves. Throughout the paper, we have added more discussion about the differential roles of gap junctions and nAChR on Stage 1 waves.

– The first part of the Results could be better organized. It is not clear to me if the authors try to explain the relevant methods of analysis, illustrate the different types of activity they found (showing individual examples), or describe activity showing population data. In the second paragraph, the authors describe how they acquired the data, this is fine. In the third paragraph, the authors explain how they analyzed the movies and some properties of the transients detected per ROI and how ROIs participate in waves (although there is not, beforehand, a clear definition of what a wave is). Here, some clarity is needed when the authors use the terms "transients", "events", and "waves". Up to this point, the authors refer to figure 1 where they show data from sample experiments. But, in the last paragraph of this section, they move to population data (and jump suddenly to panels of figure 3). Moreover, they focused on large propagating events, though it seems that sometimes they refer to all the propagating events, and present a summary in Table 1, where surprisingly small waves appeared for the first time without being mentioned in the text. In sum, it would be helpful for the reader if the authors reorganize the text of this section, try to set a clear scope for it, and present the data as smoothly and clearly as possible.

We have modified significantly the first part of the results to address these concerns. Specifically, we:

1) Rearranged the presentation of the results so that the flow is better. Figure 1 has been rearranged so that captions are called in order and there is no jump to Figure 3.

2) Increased clarity by using only the two terms “transients” and “waves”, and removing the word “event”. Transients and waves are both clearly defined in revised text page 4.

– Are waves detected by looking at the "% Active" plots, by visual inspection of the movies or by an algorithm? In the first case, what happens if there are two or more waves running simultaneously? It would help the reader if the authors explain how waves were detected. If wave detection is manual or visual, why the authors did not use automatic detection? In figure S2, it seems that there are a lot of waves per experiment, too many to be quantified manually or by eye.

We have added information in the methods that describe how waves are detected. Specifically, the waves are automatically detected using a peakfind algorithm in MATLAB on the % Active traces.

– I suggest modifying the title to better fit the data. In the first half of the manuscript, the authors investigate the spatiotemporal properties and pharmacological profile of embryonic retinal waves. Using this information, the authors propose a model for the initiation and propagation of stage 1 waves (figure 6), but none of the cell types of the model were directly assessed in the manuscript. Regarding the ipRGCs, the authors nicely show that these cells participate in the network activity at early developmental stages; however, the data does not show the mechanism related to ipRGCs. Strictly, and in my opinion, the cellular mechanisms are not really established here.

We modified the title to: Circuit mechanisms underlying embryonic retinal waves

– Why is the frequency of waves in the Hex control (Figure 2B) half the frequency of the other controls?

There was variance across preparations. However, we have conducted additional experiments and found that the frequency of waves within the same range as other experiments. The new data is incorporated into part of Revised Figure 2.

– The authors should include the mean duration of the waves/events in the description of the different types of activity.

Duration is convolved with velocity, area and kinetics of the calcium dye. Therefore we restrict our description in the paper to area and velocity. It is unclear to us why this parameter is needed and so we have not added it.

– The authors claim that stage 1 retinal waves persist in WT retinas treated with DHβE. It is true that some activity remains upon DhβE, but, to me, the raster plot looks different than the WT (figure 2D). The statistical test yields no significant differences, but the p-value is amazingly borderline p = 0.05. This means that more experiments are needed to raise robust conclusions. Moreover, since DhβE (figure 2E) reduce the area of the events from 90% to 10% … is it still possible to call them stage 1 retinal waves? In sum, when the authors claim that waves persist, it suggests that normal stage 1 waves were observed in the DhβE-treated retinas; however, what remains looks like something different.

We clarified how we describe this result (bottom of page 5).

“However, in contrast to Stage 2 wave, some waves persisted in the presence of DhβE, but they recruited fewer neurons and therefore had smaller areas (Figure 2B)”

– As stage 1 waves are also affected in β2-nAChR KO mice, which is the "standard" model that has been widely used to assess the function of stage 2 waves, this is a good opportunity for the authors to discuss to what extent the conclusions raised using this mouse in the previous literature needs to be re-interpreted.

We revised the discussion significantly. We highlight page 9 – the text is reproduced here for reviewers convenience:

“The different sensitivity to nAChR antagonists between Stage 1 and 2 waves is highlighted in the patterns of retinal waves in β2-nAChR-KO mice. β2-nAChR-KO mice have significantly reduced Stage 2 cholinergic waves (Bansal et al., 2000; Burbridge et al., 2014; Xu et al., 2015, 2016) and as such have served as the canonical model for studying the role of Stage 2 cholinergic waves in eye-specific segregation, retinotopic maps, retinal and collicular direction selectivity, and in the optokinetic reflex (Arroyo and Feller, 2016; Grubb et al., 2003; Thompson et al., 2017; Tiriac et al., 2022; Wang et al., 2009). In contrast, Stage 1 waves in the β2-nAChR-KO mice persist, albeit they propagate with slower speed and cover smaller areas of the retina (Figure 3). Despite this difference in spatiotemporal properties, both Stage 1 and Stage 2 waves that persist in the β2-nAChR-KO mice are blocked by gap junction receptor antagonist rather than blockers of fast neurotransmitter receptors (Kirkby and Feller, 2013). Recent evidence suggests that embryonic activity throughout the developing sensory system may influence many aspects of visual systems development (Martini et al., 2021; Moreno-Juan et al., 2022). Whether the various visual system phenotypes observed in β2-nAChR-KO mice can be attributed in part to reduced Stage 1 waves remains to be determined.”

Additional comments (Introduction is page 1)– The term "events" is used both to indicate the activation of an ROI and the correlated activation of multiple ROIs (as in waves). It would be easier for the reader if the authors differentiate them. Examples of the first case can be found in the third paragraph of the results: (i) "… traces for each ROI were rasterized based on an event detection algorithm …" or (ii) "We first computed the time between spontaneous events by measuring the inter-event interval (IEI) for each ROI and …". Examples of the second case can be found in the fourth paragraph: (i) "In addition to propagating events …" or (ii) "These events were like those described previously in (Bansal et al., 2000)". In addition, this disambiguation will clarify what they quantify in the "Events/min" plots. In sum, the authors should somehow clearly differentiate the activation of single ROIs from population activity. Perhaps, "transients" is a good option for the activation of single ROIs. It is not clear from the context which class of events is referred to by "These".

We clarified the wording of transients, events, and waves and defined these terms on page 4.

– Why are waves divided into small and large events? This subdivision is not used in any of the quantifications that follow figure 1. In the same line, the authors adequately classified the events in propagating and non-propagating but we don`t know what happens with the latter in the different experimental conditions analyzed. It would be interesting to know whether the properties of non-propagating events are changed or remain the same upon pharmacological or genetic manipulations.I think that the description of how to classify large waves is not accurate. In Methods: "Large propagating waves were identified as regions of correlated calcium activity with no fixed areas and with wavefronts covering up to 90% of the retina". I guess that the authors meant wavefronts covering areas larger than 25% of the retina, which is the upper limit for small waves. In addition, the authors classify the events as fixed or non-fixed. However, it is not clear to the reader what "fixed" means.

We no longer make the distinction between small and large waves because there is a continuous distribution of wave areas. See Figure 1 supplement 1.

– The number of asterisks used to indicate statistical significance is not consistent (see Figures 2 and 3, for example).

We fixed this to make it consistent across figures.

– The algorithm that detects calcium transients within ROIs should be described or referenced.

We added this information to the revised methods.

– Why does the threshold for detection vary from 15% to 50% depending on the experiment? In addition, these thresholds seem very high. Why not use n-times as the SD of the baseline? Is it related to the signal-to-noise ratio?

See new figure 1B. Our signal-to-noise is very good for these recordings, S Thresholds varied with dye that is used. In all cases, thresholds > 10 SD were used.

– I have some comments that may help to refine the Abstract. In lines 2-3, the authors should mention the species. In line 5, as far as I know, stage 1 waves are also observed at least at P0-P1, it is thus not completely correct to use "embryonic retinal waves" as a synonym for stage 1 waves. Perhaps "the earliest pattern of retinal waves" will do it. In line 7, it is not clear to me what the authors mean by "finite". And finally, by reading only the Abstract, it is not evident which could be the relationship between stage 1 waves and the distribution of ipRGCs. Perhaps, the authors should briefly mention here what normally happens to this subpopulation during perinatal development.

We have really avoided P0 recordings here and therefore cannot make much of a comment about the transition between Stage 1 and Stage. But I do think it is fair to call embryonic waves “Stage 1”.

We have eliminated the word “finite”

We have added.. “which undergo a significant amount of cell death” too

– Page 1, line 7. Typo: vision instead of "vison". Page 2, line 26. Typo: characterized instead of "characterize".

These have been corrected.

– Page 2, lines 29-30. The details of FOV and resolution should go to Methods. The authors emphasize here that the field of view could cover the whole retina while preserving cellular (not subcellular) resolution, this is clear even without the details, increasing the readability of the paragraph. Please, check that "FOV" is spelled out before being abbreviated in the main text.

We clarified the text but kept some of this methodology wording in the Results sections so that the distinction between transients and waves is clear.

– Do the authors think that "complex" is a good descriptor for the spatiotemporal properties of stage 1 waves (as stated in the first subheading of Results)?

We have removed the word complex and changed the first subheading to

Macroscope imaging reveals the spatiotemporal properties of Stage 1retinal waves.

– I think that the initial paragraph of the first part of Results, about the cell types in the developing retina, does not have a clear link to the following paragraphs of the section. Perhaps, this information fits better in the Introduction. Further, I believe that Bansal and colleagues is not a convenient reference for the last sentence where gap junctions are proposed to be a primary substrate of stage 1 waves. There is only one mention of gap junctions in this publication. The authors may add here the Wong and colleagues (1998)*.*

Our hope here was to motivate the specific pharmacological manipulations we conducted and to better describe what we mean by “immature state of retinal circuits”. We have modified text to hopefully make that clearer. We added the Wong and colleagues, 2018 reference and removed the Bansal reference.

– Page 3, line 5. I think that it is more appropriate to give the average area of these events in square microns rather than in the number of ROIs occupied by them.

We have made this change.

– Page 3, line 23. "E16-18 mouse pups". Better replace mouse pups with "mice" or "mouse embryos".

We have made this change.

– Page 4, line 11. Figure 3D is not the correct panel for these data.

We have made this correction.

– Page 5, line 22. "We observed a dramatic decrease in the density of ipRGCs in WT retinas from P1 to P7 (Figures 5B, C), …". Although differences are evident, this statement needs statistical analysis.

We have added p values for the change in ipRGC density from P1 to P7 in both WT and β2Kos.

– The authors claim that IEIs are longer in beta2-nAChR-KO than in WT retinas (figure 3E). A statistical analysis should be provided here that directly compares these two populations (Black and pink violins in the panel).

We have added this analysis in revised Figures 3D and E.

– In the caption of figure 1B, I guess that "an E17" means "an E17 retina".

We have modified text accordingly.

– In the caption of figure 2, p-values do not correspond to the panels.

We have added p-values to all figures and made this correction in Figure 2.

– It is convenient that the authors add the maximum number of ROIs represented to the X-axis of raster plots (for instance, 1 at the bottom and say 500 at the top of the axis).

We have modified the figure accordingly.

– In Figures 2F and H, the authors show time-series data points; however, there are points without lines in the graphs.

We have removed these points.

– Rows in Table 1 are individual experiments. Please, add this information to the Table.

We added this information to the caption.

– In the speed quantification, the authors explain that the first point of the wavefront is selected at random. I understand then that the speed value for a wave will change every time that the quantification is repeated (because wavefronts do not seem to propagate homogeneously). Is this correct? Or, are measurements repeated many times for every wave and a mean calculated? Also, when the second point is selected, it is not clear to me what "parallel to the direction of the first point" means. This section will benefit from a better explanation of the quantification method. Is this method standard or comparable to other methods that have been used in previous publications?

We clarified the description of the wave speed measurements. This method was used to give an average measurement to wavefront propagation. We implemented several more complex algorithms based on location motion along all edges of wavefronts but given the complexity of Stage 1 waves, this was not productive. Rather we wanted to capture a more macroscopic assessment of wave propagation to compare between WT and β2-nAChR-KO. This approach led to similar Stage 2 velocities reported in previous publications. To describe the method we have added a new Figure 1 Supplemental Figure 2.

– Could the authors state in each figure the number of independent experimental units (N): how many retinas from how many mice (for example, information is incomplete in figure 3 and missing in figure 5)?

We have now added this to the Figure captions for Figures 3 and 5.

– "Cook and Bekker, Physiology, 2009" is not listed in the reference list. "Emanuel and Do, 2015" is not listed in the reference list.

We have added both of these references.

[Editors’ note: further revisions were suggested prior to acceptance, as described below.]

The paper has improved in revision, and most of the original issues have been nicely handled. A few issues remain or have been introduced in the revisions.MFA. The additional discussion of MFA is helpful though, as noted in the paper, ultimately not entirely conclusive. Did you try to wash out the effects of MFA in any of the experiments described here? Even a partial washout would be helpful. I would consider moving the whole cell recordings on lines 155-158 to the Methods since the variability limited the conclusions that can be drawn from them. It does look like K channel activity decreased systematically in MFA, which might be noted.

We have made the changes you suggested – moving the results based on whole cell to the methods but leaving the summary in the main text.

For this study, we have not systemically checked for a washout of MFA. We do not see any signs of general health decline with the calcium imaging which manifests itself by a steady increase in baseline of the fluorescence (which in the past we have seen with higher concentrations of MFA). For whole cell recording, in two cells, we saw partial recovery of the voltage-gated potassium conductance after 15 minutes of rinse (after 45 minutes in 50 µM MFA) – given washout issues this seems not useful to add to paper. We also do not see big changes in resting potentials of cells or voltage-gated Na conductances. It is important to note that in the past we have found 50 µM MFA was sufficient to block gap junction coupling as assayed with spikelets (Arroyo et al., 2016, Figure 7; Caval-Holme et al. 2019; Figure 4).

The motivation for the measurements in the knockout animals is clearer now. The conclusion that compensation causes gap junctions to contribute more strongly to waves in the knockout mice could be highlighted more (it currently comes in the middle of the paragraph starting on line 180). I would consider adding that conclusion to the abstract or introduction and making it clearer in the results.

We cannot conclude there was compensation – rather gap junctions are the remaining circuit mechanisms for propagating waves in the knockout mice. We have made modifications to Lines 180 and 280 to clarify this point.

The abstract is quite long and detailed. I would suggest removing some of the sentences about the experimental approach to the final paragraph of the introduction, and focusing the abstract more on conclusions to make it more generally and easily accessible.

We have streamlined the abstract.

ipRGC development and waves. Since stage 1 waves are modified but not eliminated in the knockouts, some of the conclusions about waves and ipRGC development seem too strong (e.g. lines 250-251, line 265, lines 283-284). The last sentence of the Discussion strikes me as a balanced way to handle this issue, so some of that could come into the earlier statements.

page 250-251 – cell death primarily occurs P1 – P7 (Stage 2 waves) and during this time waves are nearly absent in b2-nAChR KOs. Cell differentiation occurs during Stage 1 waves. That said, we have modified the sentence but not weakened the conclusion too much. Specifically, we added the following sentence to address your point:

“Whether elimination of all spontaneous activity affects these processes remains to be determined.”